# Unlocking The Power Of Layer-By-Layer Training And Fine-Tuning

## Abstract

Layer-wise (LW) training of deep neural networks has long been associated with memory and parallelism advantages, yet it suffers from information degradation and poor convergence in deep architectures. Recent work attributes these issues to the loss of input information and the lack of layer-role differentiation, as measured by the Hilbert-Schmidt Independence Criterion (HSIC).

In this paper, we present a novel algorithm that enables full end-to-end training of ResNet-18/ResNet-50 and end-to-end fine-tuning of Large Language Models (LLMs) using a modified LW approach, while minimizing performance degradation. Our fundamental contribution lies in the discovery that strategically reintroducing the final layers during LW training mitigates the convergence degradation typically observed during LW when compared to conventional end-to-end fine-tuning.

We introduce Segmented Propagation (SegProp), a training paradigm that seamlessly integrates the computational efficiency of LW optimization with the representational power of global supervision. Quantitative results demonstrate substantial improvements in convergence compared to standard LW fine-tuning of LLMs and compared to LW training of ResNet-18/ResNet-50. SegProp improves ResNet-50 accuracy on CIFAR-10 from 90.0% (LW) to 94.3%, approaching E2E training at 95.5%. On ResNet-18, SegProp improves CIFAR-10 accuracy from 93.7% (LW) to 95.2%, closely matching E2E at 95.5%. On Mistral-Nemo-Instruct-2407, SegProp segmented fine-tuning matches E2E MMLU (5-shot) performance (69.3%), and for Llama3.1-8B-Instruct it achieves 78.9% on Winogrande (5-shot), closely matching E2E fine-tuning at 79.1%.

## 1 Introduction

Training modern large language models (LLMs) and deep neural networks is increasingly limited by memory and communication. For billion-parameter LLMs, end-to-end (E2E) backpropagation often exceeds single-GPU memory even with techniques such as activation checkpointing and quantization (HuggingFace, 2023), pushing practitioners toward pipeline parallelism (Wang et al., 2025).

LW training, where blocks are optimized using local losses, reduces peak activation memory and offers flexible parallel scheduling (Bengio et al., 2006; Hinton et al., 2006). Yet it consistently underperforms E2E optimization in deep models (Sakamoto & Sato, 2024). Prior work links this gap to information loss: local objectives encourage intermediate layers to discard input information that remains useful for the final prediction (Sakamoto & Sato, 2024; Wang et al., 2021). HSIC-based analyses further show that representations become progressively less informative with depth (Tishby et al., 2000; Tishby & Zaslavsky, 2015). A central issue is the lack of persistent global supervision: each segment is trained to satisfy its own auxiliary head rather than the model's final task.

We argue that segmentation itself is not the core problem; the missing piece is a persistent global target. We introduce Segmented Propagation (SegProp), a training paradigm that keeps a small *global head*—the model's final layers and task-specific head—active and trainable throughout training. Earlier segments are optimized one at a time, but always under a single task-level loss computed by this shared head. Lightweight training-only adapters bridge intermediate features to the head when needed and are removed at inference. This preserves global task information at every depth while retaining much of the memory and parallelism benefit of segmented training.

Our goal is to reduce peak training-time memory and optimizer state—enabling larger micro-batches or deeper models on the same hardware—without sacrificing E2E-level performance. Concretely, we:

- Propose **Segmented Propagation (SegProp)**, which combines the efficiency of segmented/LW training with the representational power of global supervision by maintaining a persistent global head and a single task-level loss for all segments.

- Introduce a practical **two-stage procedure**: (i) train a base prefix jointly with the global head; (ii) iteratively train intermediate segments while keeping the global head active, using lightweight adapters that are discarded at inference.

- Show that SegProp substantially narrows the gap between LW and E2E training on CNNs and fine-tuning on LLMs: on CIFAR-10, SegProp improves ResNet-50 accuracy from 90.0% (LW) to 94.3%, approaching E2E at 95.5%; for Mistral-Nemo-Instruct-2407 and Llama3.1-8B-Instruct, SegProp segmented fine-tuning matches or nearly matches E2E performance on MMLU and Winogrande.

Figure 1 contrasts standard LW training, E2E backpropagation, and SegProp.

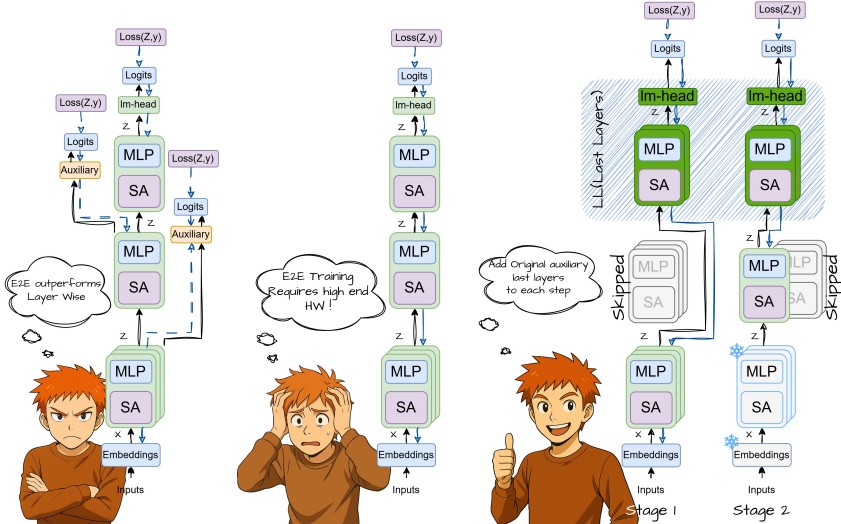

Figure 1: From left to right: layer-wise (LW), End to End (E2E), Segmented Propagation (SegProp).

## 2 RELATED WORK

### 2.1 LAYER-WISE TRAINING AND INFORMATION BOTTLENECK

Layer-wise (LW) training was originally proposed to address the credit assignment problem and to provide improved initialization for deep networks (Bengio et al., 2006; Hinton et al., 2006). More recently, empirical studies indicate that LW training can suffer from information degradation as depth increases, often quantified using the Hilbert–Schmidt Independence Criterion (HSIC) as a proxy for mutual information. Such degradation is associated with poorer generalization and diminishing accuracy gains in deeper models (Sakamoto & Sato, 2024; Wang et al., 2021). The information bottleneck principle (Tishby et al., 2000; Tishby & Zaslavsky, 2015) offers a useful lens on these dynamics: end-to-end (E2E) optimization tends to compress intermediate representations while preserving task-relevant information in the final layers.

A central limitation remains: standard LW training lacks global supervision. As a result, intermediate representations may discard input information that is important for the final prediction task, leading to weak convergence and limited accuracy improvements with increased depth.

We address these limitations by rethinking the role of the final layers in segmented optimization (see Figure 1). Specifically, we propose *Segmented Propagation (SegProp)*, which reintroduces the final layers during LW training as a shared global head. This design restores task-level information flow to earlier segments while retaining the efficiency benefits of segmented optimization. By providing consistent global supervision throughout training, SegProp encourages intermediate layers to learn general, transferable features rather than prematurely specializing for classification, yielding competitive accuracy with improved memory efficiency.

## 2.2 BIOLOGICALLY PLAUSIBLE AND MODULAR TRAINING

Alternatives to backpropagation, such as Hebbian learning (Hebb, 1949), reservoir computing (Bianchi et al., 2020), and signal propagation (Kohan et al., 2022), aim to reduce memory usage and computational cost by localizing learning. Modular and block-wise training strategies (Belilovsky et al., 2018; Gomez et al., 2022) seek to balance parallelism and accuracy, but they often rely on backpropagation within blocks. The Forward-Forward algorithm (Hinton, 2022) enforces distinct roles for each layer, but it still suffers from information loss as depth increases.

## 2.3 ACTIVATION CHECKPOINTING AND MEMORY-EFFICIENT TRAINING

Checkpointing reduces peak memory by storing a subset of activations and recomputing the rest during the backward pass, trading compute for memory (Chen et al., 2016; He & Yu, 2023; Purandare et al., 2023; Korthikanti et al., 2022). It is a key tool for scaling deep models under constrained resources (Sakamoto & Sato, 2024). Activation checkpointing (AC) and selective AC (SAC) are standard: SAC applies checkpointing only to critical layers to balance recomputation overhead against savings. While effective, recomputation can add latency and bandwidth pressure, especially in distributed or multi-GPU settings.

**SegProp memory model.** In SegProp, only the active segment and the shared global head require gradients and optimizer state; all other segments run forward-only, reducing peak activation memory and optimizer footprint while preserving a single task-level loss via the persistent head.

**Snapshot checkpointing (SnapCheck).** Complementary to AC/SAC, SnapCheck caches detached activations from the frozen prefix and reuses them across iterations, reducing redundant forward compute without increasing peak memory. It integrates naturally with SegProp and is used only during training.

## 3 SEGMENTED PROPAGATION (SEGPROP)

### 3.1 PROBLEM SETTING

As previously noted, end-to-end (E2E) training of large models demands significant GPU memory: it must retain weights, activations, optimizer states, gradients, and other intermediates across the entire depth, leading to high peak memory use and extended runtimes.

To mitigate this burden, layer-wise (LW) training updates one layer or sub-block at a time rather than performing full-depth backpropagation (Bengio et al., 2006; Hinton et al., 2006). By restricting gradient traversal to the *active* segment and running earlier/later segments forward-only, LW substantially reduces the peak activation footprint and compute per step. However, because supervision is local to the active segment (often via an auxiliary head), LW can suffer information loss and suboptimal performance: intermediate representations may be insufficiently informative for the final task (Sakamoto & Sato, 2024). In agreement with prior results on convolutional architectures (ResNet-18/50, VGG11) showing sizeable accuracy degradation under LW (Sakamoto & Sato, 2024), our experiments on a Transformer-based architecture fine-tuning reveal an even more severe performance drop under MMLU and HumanEval+ (HE+) (Liu et al., 2023).

E2E training remains the standard approach: the network runs forward to produce outputs, the loss is computed, and stochastic gradient descent (SGD) applies the chain rule to propagate gradients through *all* layers in reverse order.

LW training, in contrast, proceeds stage-wise. For the first segment, the model runs forward through that segment and a small auxiliary module that aligns its output to the task objective; the segment is optimized against a local loss while the rest of the network remains frozen. Subsequent stages repeat this pattern for each segment: the network runs up to the current segment, a geometry-matching auxiliary module produces the supervision signal, the loss is computed, and only the active segment is updated. Auxiliary modules are training-only and discarded at inference.

We propose Segmented Propagation (SegProp): a method grounded in LW's segmented optimization but enhanced by reintroducing the final layers during every stage. Concretely, SegProp maintains a *shared global head*—the final few layers that produce the network's output—throughout training. The active segment is trained against a *single task-level loss* produced by this shared head (fed via lightweight adapters to match geometry when needed). This restores global supervision and aligns representations across stages, helping recover information loss. Empirically, SegProp achieves competitive accuracy with E2E fine-tuning of LLMs and training from scratch for ResNet architectures, while preserving most of the efficiency benefits of segmented training.

### 3.2 Two-Stage SegProp Strategy

SegProp employs a two-stage training strategy designed to balance memory efficiency with strong supervision:

1. **Joint training of a base prefix with the last layers.** Select a prefix of the model (base layers) and train it jointly with a small set of final layers (the *last-layers* module) to establish a strong task-level signal early. Intermediate layers are skipped to reduce compute and memory. Gradients flow through the base prefix and last layers only; earlier/later layers are non-trainable in this stage.

2. **Iterative layer-wise training of intermediate layers with global supervision.** Train each intermediate layer individually alongside the last-layers module, which provides a consistent, task-level loss across stages. Previously trained layers remain frozen; only the current target layer and the last layers receive updates. After a layer is trained, its weights are committed (frozen) for the subsequent iteration. Lightweight adapters (training-only) are used as needed to match geometry between the current layer's output and the last-layers input; these adapters are discarded at inference.

### 3.3 Formalization

We first describe SegProp in an architecture-agnostic way for a generic deep network, and then instantiate it for Transformers and ResNets in the experiments.

**Model and segmentation.** Let $\{(x_i, y_i)\}_{i=1}^{m}$ be a dataset with inputs $x_i \in \mathcal{X}$ and targets $y_i \in \mathcal{Y}$, and let $f_\theta : \mathcal{X} \to \mathcal{Y}$ be a deep network with parameters $\theta$. We view $f_\theta$ as a composition of $n$ trainable blocks (layers or segments):

$$f_\theta(x) = h \circ f_{n-1} \circ f_{n-2} \circ \cdots \circ f_0(x), \tag{3.1}$$

where $f_j$ denotes the $j$-th block (e.g., a ResNet stage or a Transformer decoder layer) and $h$ is the final task-specific head (e.g., classifier or LM head).

For compactness, define the half-open composition

$$F_{a:b}(x) := f_{b-1} \circ f_{b-2} \circ \cdots \circ f_a(x), \quad 0 \le a < b \le n, \tag{3.2}$$

with the convention $F_{a:a}(x) = x$.

We choose:

- a *base prefix* depth $p \in \{0, \ldots, n-1\}$,
- a *last-layers* size $r \in \{1, \ldots, n-p\}$.

The base prefix and last-layers module are then

$$f^{[0:p]}(x) := F_{0:p}(x), \tag{3.3}$$

$$\mathcal{LL}^{(r)}(z) := h \circ F_{n-r:n}(z), \tag{3.4}$$

where $F_{n-r:n}$ denotes the composition of the last $r$ blocks preceding the head $h$. Intuitively, $\mathcal{LL}^{(r)}$ is a small *global head* consisting of the head $h$ and the last $r$ blocks of the network. All remaining blocks, with indices $j \in \{p, \dots, n-r-1\}$, are treated as *intermediate* and will be trained iteratively.

We denote by $\mathrm{StopGrad}(\cdot)$ an operator that prevents gradient propagation through its argument (e.g., `detach()` in PyTorch), ensuring that gradients do not flow into earlier segments when we train a given block.

**Stage 1: joint training of base prefix and global head.** In Stage 1, we train the base prefix $f^{[0:p]}$ jointly with the last-layers module $\mathcal{LL}^{(r)}$, while skipping intermediate blocks.

Given a minibatch $\mathcal{B} = \{(x^{(b)}, y^{(b)})\}_{b=1}^{|\mathcal{B}|}$, the forward pass is

$$z^{(b)} = f^{[0:p]}(x^{(b)}), \tag{3.5}$$

$$\tilde{z}^{(b)} = A_p(z^{(b)}), \quad \text{(optional adapter; identity if not needed)} \tag{3.6}$$

$$\hat{y}^{(b)} = \mathcal{LL}^{(r)}(\tilde{z}^{(b)}), \tag{3.7}$$

with task loss

$$\mathcal{L}_1 = \frac{1}{|\mathcal{B}|} \sum_{b=1}^{|\mathcal{B}|} \ell(\hat{y}^{(b)}, y^{(b)}), \tag{3.8}$$

where $\ell$ is the end-task objective (e.g., cross-entropy). Backpropagation updates only the parameters of $\{f^{[0:p]}, \mathcal{LL}^{(r)}, A_p\}$; all other blocks are frozen. After convergence, we *commit* (freeze) the base prefix $f^{[0:p]}$, while keeping $\mathcal{LL}^{(r)}$ active and trainable for Stage 2.

**Stage 2: iterative training of intermediate blocks with global supervision.** In Stage 2, we train the intermediate blocks one at a time under a *single* task-level loss provided by $\mathcal{LL}^{(r)}$. Let $\hat{p}$ index the current block in $\{p, \dots, n-r-1\}$. For minibatch $\mathcal{B}$, the forward pass is

$$z_{\hat{p}}^{(b)} = f_{\hat{p}}\Big(\mathrm{StopGrad}\big(F_{0:\hat{p}}(x^{(b)})\big)\Big), \tag{3.9}$$

$$\tilde{z}_{\hat{p}}^{(b)} = A_{\hat{p}}(z_{\hat{p}}^{(b)}), \quad \text{(adapter; identity if not needed)} \tag{3.10}$$

$$\hat{y}^{(b)} = \mathcal{LL}^{(r)}(\tilde{z}_{\hat{p}}^{(b)}), \tag{3.11}$$

and the loss is

$$\mathcal{L}_2 = \frac{1}{|\mathcal{B}|} \sum_{b=1}^{|\mathcal{B}|} \ell(\hat{y}^{(b)}, y^{(b)}). \tag{3.12}$$

Here, gradients are applied only to $\{f_{\hat{p}}, \mathcal{LL}^{(r)}, A_{\hat{p}}\}$; all blocks with indices $< \hat{p}$ have been committed and remain frozen, and blocks with indices $> \hat{p}$ are not part of the computation graph at this stage. After training block $\hat{p}$ to convergence, we commit $f_{\hat{p}}$, discard $A_{\hat{p}}$ (training-only), and move to the next block.

**Efficiency and global supervision.** At any time, SegProp activates gradients only through the base prefix (Stage 1) or a single intermediate block (Stage 2) plus the shared global head. This reduces peak activation memory and optimizer state relative to end-to-end backpropagation, where gradients must flow through all $n$ blocks. At the same time, the persistent last-layers module $\mathcal{LL}^{(r)}$ enforces a *single* task-level objective across all segments.

In our experiments, we instantiate this generic scheme with:

- Convolutional backbones (ResNet-18/50), where $f_j$ are ResNet stages and $\mathcal{LL}^{(r)}$ reuses the final convolutional layers and classifier (Appendix B); and

- Transformer-based LLMs, where $f_j$ are decoder layers and $\mathcal{LL}^{(r)}$ consists of the last few decoder layers and the LM head (Appendix C).

### 3.4 SNAPSHOT CHECKPOINTING (SNAPCHECK)

To avoid redundant computation during Stage 2, we introduce *Snapshot Checkpointing* (SnapCheck), a cache of frozen-prefix activations that can be reused across layer iterations. After computing the committed prefix output (Eq. (3.13)), for a minibatch $\mathcal{B} = \{(x^{(b)}, y^{(b)})\}_{b=1}^{|\mathcal{B}|}$ we form

$$z^{(b)} \;=\; f^{[0:p]}(x^{(b)}), \qquad \forall b \in \{1, \ldots, |\mathcal{B}|\}, \tag{3.13}$$

and store *detached* snapshots $\bar{z}^{(b)} := \texttt{StopGrad}(z^{(b)})$ in a memory-efficient buffer $\mathcal{S}$, indexed by the prefix depth $p$ and a minibatch identifier (e.g., dataloader index, seed). When fine-tuning a subsequent layer $f_{\hat{p}}$, instead of recomputing $f^{[0:p]}(x^{(b)})$ on every step, we retrieve the cached activation:

$$z^{(b)} \;\leftarrow\; \mathcal{S}[p, \texttt{batch\_id}], \tag{3.14}$$

falling back to on-the-fly computation and insertion if the snapshot is missing.

Because $f^{[0:p]}$ is committed (frozen) during Stage 2, these snapshots remain valid across iterations, eliminating repeated evaluation of the prefix. This targets reduction of per-step compute and wall-clock time, especially when the committed prefix is deep ($p \gg 0$). SnapCheck is complementary to activation checkpointing (AC/SAC): AC reduces the *activation memory* of the trainable suffix, while SnapCheck reduces *forward compute* by reusing frozen-prefix outputs. Snapshots are training-only; inference uses the standard forward h $\circ$ $F_{0:n}$ without snapshots. In practice, SnapCheck stores detached prefix activations per batch index and reuses them across iterations when only deeper layers are being updated, so the cost of the frozen prefix is paid once per batch per prefix depth.

A step-by-step pseudocode description of Segmented Propagation Stochastic Gradient Descent (SegProp-SGD), including the handling of adapters, snapshot checkpointing, and freezing policies, is provided in Algorithm 1 in Appendix A. In practice, this algorithm implements the two-stage procedure described above: (i) joint training of a base prefix with the last-layers module, followed by (ii) iterative training of intermediate layers under a persistent global head and a single task-level loss.

## 4 RESULTS AND ANALYSIS

### 4.1 SEGPROP TRAINING FOR RESNET-18 ON CIFAR-10

**Overview.** ResNet-18 is trained with SegProp by splitting the backbone into segments optimized in stages while retaining a shared global head throughout training. The head is always in the loss path and updated at every stage; training-only adapters map segment outputs to the head's input geometry.

#### 4.1.1 BACKBONE SEGMENTATION AND GLOBAL HEAD

We use an ImageNet-style ResNet-18 with 224×224 inputs, partitioned into four segments (stem + layer1, layer2, layer3, and part of layer4). A small global head (final convolutional sub-layer plus 2-layer MLP classifier) remains shared across all stages, while lightweight training-only adapters map each segment's output into the head's input geometry (details in Appendix B).

In the notation of Section 3.3, ResNet-18 is decomposed into four segments $f_0, \ldots, f_3$ (stem + layer1, layer2, layer3, part of layer4), and the global head $\mathcal{LL}^{(r)}$ consists of the remaining layer4 convolution and the final MLP classifier.

#### 4.1.2 LAYER-WISE TRAINING SETTING FOR RESNET-18

For LW training the backbone is split into four segments (seg1–seg4), each trained sequentially with its own auxiliary classifier, followed by a final 'fc' stage that trains only the global classifier with the backbone frozen. At stage k (1–4), all earlier segments are frozen and only the current segment and its auxiliary head are updated. Auxiliaries are lightweight Conv–BN–ReLU adapters that normalize intermediate features to a fixed 512×7×7 resolution, followed by global average pooling and a linear classifier. Full architectural details are given in Appendix B.

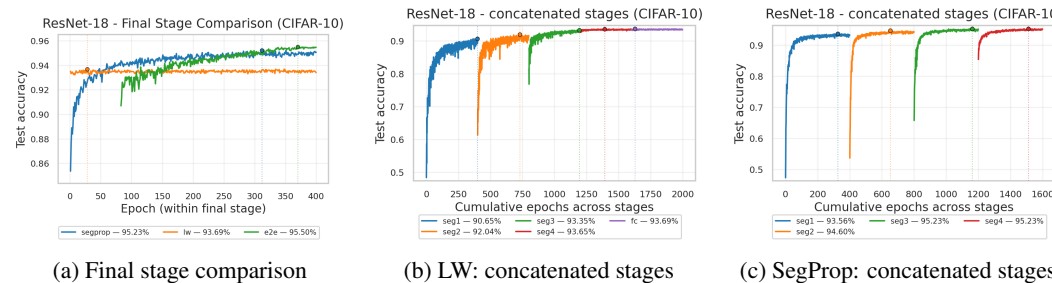

(a) Final stage comparison     (b) LW: concatenated stages     (c) SegProp: concatenated stages

Figure 2: ResNet-18 on CIFAR-10. (a) Final-stage trajectories comparing SegProp (best 95.23%), LW (best 93.69%), and E2E (best 95.50%). (b) LW training across stages. (c) SegProp training across stages. Dots denote best-epoch checkpoints.

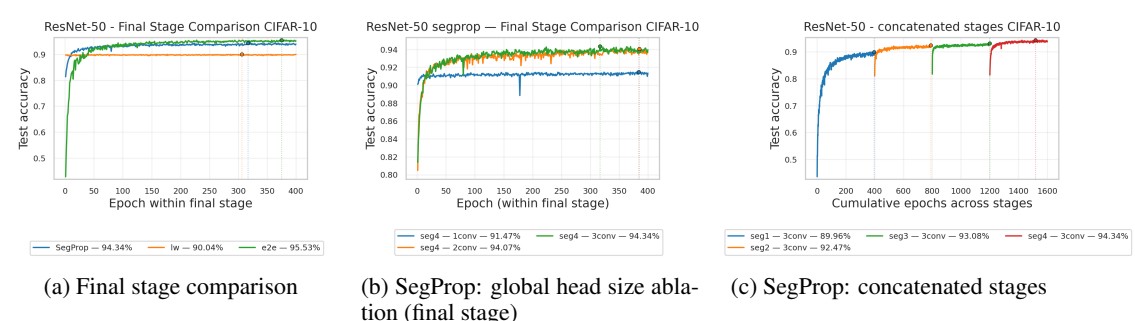

(a) Final stage comparison     (b) SegProp: global head size ablation (final stage)     (c) SegProp: concatenated stages

Figure 3: ResNet-50 on CIFAR-10. (a) Final-stage trajectories comparing SegProp (best 94.34%), LW (best 90.04%), and E2E (best 95.53%). (b) SegProp global head size ablation in the final stage: 1 conv 91.47%, 2 conv 94.07%, 3 conv 94.34%. (c) SegProp training across stages: seg1 89.96%, seg2 92.47%, seg3 93.08%, seg4 94.34%. Dots denote best-epoch checkpoints.

### 4.1.3 RESULTS

Figure 2 summarizes ResNet-18 training on CIFAR-10 using 400 epochs (more details in Appendix B). Panel (a) compares the final-stage trajectories of SegProp, LW, and E2E, showing best test accuracy of 95.23% (SegProp), 93.69% (LW), and 95.50% (E2E). Panels (b–c) plot stage-concatenated learning curves for LW and SegProp, respectively; dots denote the best-epoch checkpoints. SegProp maintains lower peak memory compared with E2E training while closing most of the accuracy gap to E2E and clearly outperforming LW.

## 4.2 SEGPROP TRAINING FOR RESNET-50 ON CIFAR-10

Figure 3 summarizes ResNet-50 on CIFAR-10. Panel (a) compares SegProp, LW, and E2E with best test accuracy of 94.34%, 90.04%, and 95.53%, respectively. Panel (b) ablates SegProp global head size in the final stage (1conv 91.47%, 2conv 94.07%, 3conv 94.34%). Panel (c) concatenates SegProp's stages (seg1 89.96%, seg2 92.47%, seg3 93.08%, seg4 94.34%). Dots mark best-epoch checkpoints. Here, 1 conv/2 conv/3 conv denote layers derived from the last convolutional layers of the original ResNet-50 and applied as part of global head (LL).

## 4.3 MEMORY COMPARISON: SEGPROP VS END-TO-END (RESNET-50)

We now compare the peak GPU memory usage of SegProp against an E2E baseline on ResNet-50/CIFAR-10 with batch size 256 and mixed-precision training. For each regime, we instrument PyTorch's CUDA memory statistics and report the maximum of `torch.cuda.max_memory_allocated` (bytes actually held by tensors) over a steady-state epoch. Specifically, we reset statistics at the start of each epoch using `torch.cuda.reset_peak_memory_stats` and record the peak allocated memory at the end of the epoch (see Appendix B.5). For SegProp, we measure one epoch per stage; for E2E, we report

Table 1: Peak GPU memory (allocated) for ResNet-50 on CIFAR-10 with batch size 256 and AMP. SegProp runs stages **seg1–seg4** sequentially with a persistent global head; E2E uses full-depth backpropagation over all layers. Memory is reported as the maximum of `torch.cuda.max_memory_allocated` (in GB) over one epoch in steady state for each regime.

| Regime | Stage | Peak allocated (GB) |
|--------|-------|---------------------|
| E2E | full model | 11.29 |
| SegProp | seg1 | 7.09 |
| | seg2 | 4.48 |
| | seg3 | 3.47 |
| | seg4 | 2.45 |

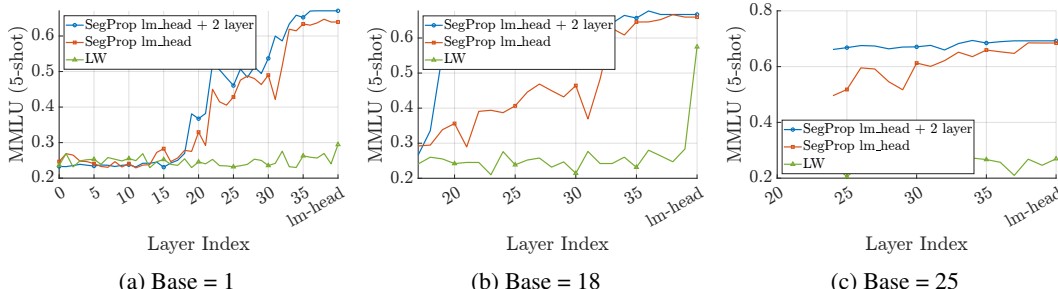

| (a) Base = 1 | (b) Base = 18 | (c) Base = 25 |

Figure 4: Nemo MMLU (5-shot) versus layer index during segmented fine-tuning for SegProp (two variants: lm-head only and lm-head + 2 layers) and layer-wise fine-tuning (LW) across three base prefixes. The final x-axis tick corresponds to the lm-head.

the steady-state peak after the initial warm-up epoch (epochs 2–4 in our run). Table 1 summarizes these peak values, converted to GB.

SegProp substantially reduces peak training-time memory compared to E2E. Even the earliest Seg-Prop stage (**seg1**), which operates on high-resolution features, uses only about 63% of the E2E peak (7.09 GB vs. 11.29 GB). Deeper SegProp stages benefit from reduced spatial resolution and thus require progressively less memory despite having more parameters: **seg4** runs at roughly 2.45 GB, a $\sim 4.6\times$ reduction in peak memory relative to E2E.

These results highlight that the primary memory cost in E2E training comes from storing activations for backpropagation through all layers. SegProp restricts gradient flow to a single segment plus the global head at any time, keeping earlier segments in forward-only mode and thereby shrinking the activation footprint while maintaining a single shared task-level objective.

In practical terms, the 1.6–4.6× reduction in peak allocated memory can be traded for larger micro-batches, deeper models, or fewer checkpointing / offloading strategies on the same hardware, while preserving accuracy close to E2E training.

### 4.4 SegProp Fine-Tuning for LLMs

Figure 4 compares Nemo MMLU (5-shot) across depth for SegProp—shown with two global heads (lm-head only and lm-head + 2 layers)—and layer-wise fine-tuning (LW).

Across base depths, SegProp consistently outperforms LW on Nemo MMLU (5-shot) (Figure 4). For a small base (1 layer), both SegProp variants quickly rise through later layers and converge near the top, while LW remains low across the stack. With deeper bases (18 and 25 layers), SegProp rapidly approaches its final performance shortly after the base boundary, and the 'lm-head + 2 layers' variant provides a small but consistent gain over lm-head-only. Overall, maintaining a global head during segmented fine-tuning robustly lifts intermediate-layer quality, and modestly extending the global head yields further gains.

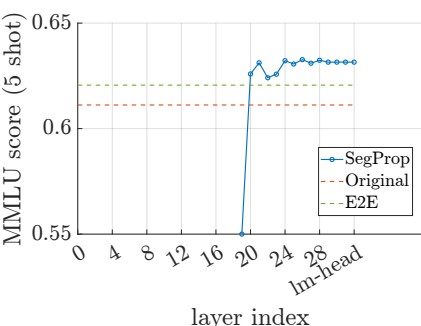
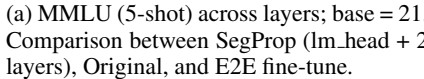
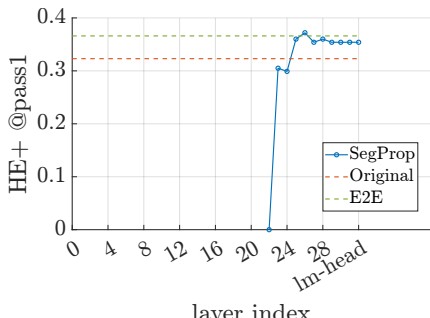

(a) MMLU (5-shot) across layers; base = 21. Comparison between SegProp (lm_head + 2 layers), Original, and E2E fine-tune.

(b) HE+ @pass1 across layers; base = 24. Comparison between SegProp (lm_head + 2 layers), Original, and E2E fine-tune.

Figure 5: Segmented fine-tuning diagnostics for Mistral-7B at two bases. Left: Base-21 MMLU (5-shot) versus layer index. Right: Base-24 coding HE+ @pass1 versus layer index. In both panels, SegProp uses a persistent global head extended by two layers (lm_head + 2) and is compared against the out-of-the-box model (Original) and an end-to-end (E2E) fine-tuned model. The final x-axis tick corresponds to the lm-head.

Table 2: Segmented fine-tuning performance across base depths for Mistral-Nemo-Instruct-2407 and Llama3.1-8B-Instruct on MMLU and Winogrande (5-shot).

| Model | Base | MMLU (5-shot) | Winogrande (5-shot) |
|---|---|---|---|
| Mistral-Nemo-Instruct-2407 | out of the box | 67.70% | 82.79% |
| | E2E Fine-Tune | 69.31% | 82.72% |
| | 1 | 67.09% | 81.29% |
| | 18 | 66.70% | 80.90% |
| | 25 | 69.38% | 81.37% |
| | 32 | 68.46% | 82.24% |
| Llama3.1-8B-Instruct | out of the box | 68.20% | 77.90% |
| | E2E Fine-Tune | 68.39% | 79.08% |
| | 1 | 67.53% | 77.19% |
| | 18 | 68.22% | 78.45% |
| | 21 | 68.45% | 78.93% |
| | 24 | 68.12% | 78.45% |

Figure 5 shows Mistral-7B segmented fine-tuning with a short persistent global head (lm_head + 2 layers). For MMLU (base = 21), SegProp scores increase sharply just beyond the base boundary, surpassing the original model and meeting the E2E fine-tuned model near the top layers. For coding (HE+, base = 24), SegProp again rises quickly at the boundary and matches the E2E baseline while remaining clearly above the original. These trends indicate that keeping a short global head active preserves task-level supervision and allows segmented fine-tuning to match full E2E fine-tuning in these settings.

As summarized in Table 2, segmented fine-tuning with an appropriately chosen base depth achieves performance that meets end-to-end (E2E) fine-tuning on both Mistral-Nemo-Instruct-2407 and Llama3.1-8B-Instruct. For Mistral-Nemo, all evaluated bases remain close to the E2E Winogrande score, and the deeper bases effectively match it. For Llama3.1-8B-Instruct, bases 18, 21, and 24 reach MMLU and Winogrande accuracy that meets the E2E fine-tune. These results confirm that SegProp can deliver E2E-level fine-tune quality without updating the entire network and are consistent with the trends observed in Figures 4 and 5.

## 5 DISCUSSION AND FUTURE WORK

Our results show that the main failure mode of conventional layer-wise (LW) training is not the use of segmented optimization itself, but the absence of a persistent global target. When every segment or layer is optimized against its own auxiliary classifier, local objectives encourage early specialization: intermediate representations become good for their local head but increasingly misaligned with the final task. Segmented Propagation (SegProp) addresses this gap by reintroducing the model's final layers as a *shared global head*, so that every segment is optimized under a consistent end-task loss.

Empirically, SegProp behaves much closer to end-to-end (E2E) training than standard LW baselines, while retaining the memory and parallelism advantages of segmented or block-wise optimization. This holds across both CNN backbones and LLM fine-tuning, suggesting that a persistent head is a robust mechanism for preserving task-relevant information in segmented training regimes.

Conceptually, SegProp sits between fully local learning and monolithic backpropagation. Like deep supervision (Belilovsky et al., 2018; Marquez et al., 2018), it injects task-level signal at intermediate depths, but it does so via the *actual* final layers rather than many separate auxiliary heads. This avoids proliferating objectives and sidesteps inference-time discrepancies, since the global head used during training is identical to the head used at test time.

SegProp is also complementary to existing memory-efficiency techniques. Its segmented optimization naturally reduces the depth over which gradients must be stored or propagated, making it compatible with activation checkpointing and selective recomputation (Chen et al., 2016; Korthikanti et al., 2022). Snapshot checkpointing (SnapCheck) can further reduce compute by caching detached prefix activations once a prefix is frozen and reusing them in later stages, which is especially attractive in multi-GPU and resource-constrained settings. Combining SegProp with parameter-efficient fine-tuning methods such as LoRA—for example, using LoRA within selected segments or within the persistent head—could further improve the trade-off between accuracy, memory, and wall-clock cost, especially in low-budget fine-tuning regimes.

**Limitations.** Our experiments focus on standard CNN backbones and 7–12B LLMs under controlled fine-tuning setups. We do not study extremely deep transformers, mixture-of-experts or multimodal architectures, or large-scale pretraining from scratch, where dynamics and system constraints may differ. We also focus on SegProp-style segmented fine-tuning and do not provide a systematic comparison to popular parameter-efficient methods such as LoRA, which offer an alternative way to reduce trainable parameters and memory footprint. Our training-only adapters are manually designed rather than learned, and we provide qualitative but not formal guarantees on when SegProp fully recovers E2E behavior.

Looking ahead, applying SegProp to larger and more heterogeneous models (e.g., MoE, multimodal, or hybrid CNN–transformer stacks) is a natural next step. Automatically designing adapters, stage schedules, or freezing policies may sharpen these trade-offs further. A more formal information-theoretic analysis, building on HSIC and information-bottleneck tools, may clarify when a single persistent head is sufficient and when richer global supervision pathways are beneficial. Overall, our findings support a simple design principle: *segmentation is not inherently at odds with global supervision*. A small, expressive global head can preserve much of the benefit of E2E training while enabling more flexible and efficient segmented optimization.

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

## A  SEGPROP-SGD ALGORITHM DETAILS

---

**Algorithm 1** Segmented Propagation Stochastic Gradient Descent (SegProp-SGD)

---

**Inputs:** Network $f(x) = h \circ f_{n-1} \circ f_{n-2} \circ \cdots \circ f_0(x)$ with $n$ blocks; base depth $p$; last-layers size $r$; loss $\ell$; optimizer(s) $\mathcal{O}$; exit criterion for Stage 1; exit criterion for block $\hat{p}$; global exit criterion

 1: **Notation:** $F_{a:b}(x) := f_{b-1} \circ f_{b-2} \circ \cdots \circ f_a(x);$ $\quad f^{[0:p]}(x) := F_{0:p}(x), p \in \{0, \ldots, n-2\};$ $\mathcal{LL}^{(r)} := h \circ F_{n-r:n}, r \in \{1, \ldots, n-p\};$ $\quad$ optional adapters $A_j$ (identity if unused)

 2: **Partition:** Decompose $f$ into non-overlapping blocks $f_0, \ldots, f_{n-1}$ (e.g., ResNet stages or Transformer layers). Always include $\mathcal{LL}^{(r)}$ during training to provide global supervision.

*Stage 1: Joint training of base prefix with $\mathcal{LL}^{(r)}$*

 3: **while** Stage 1 exit criterion not met **do**
 4: $\quad$ **for** each minibatch $(x, y)$ **do**
 5: $\quad\quad$ $z \leftarrow f^{[0:p]}(x)$
 6: $\quad\quad$ $\tilde{z} \leftarrow A_p(z)$ $\qquad\qquad\qquad$ ▷ Adapter for geometry matching; identity if not needed
 7: $\quad\quad$ $\hat{y} \leftarrow \mathcal{LL}^{(r)}(\tilde{z})$
 8: $\quad\quad$ $\mathcal{L}_1 \leftarrow \ell(\hat{y}, y)$
 9: $\quad\quad$ **Backward/Update:** backprop through $\{f^{[0:p]}, \mathcal{LL}^{(r)}, A_p\}$ and update only these parameters
10: **Commit and freeze** $f^{[0:p]}$; keep $\mathcal{LL}^{(r)}$ **trainable and persistent** for Stage 2

*Stage 2: Iterative training of intermediate blocks with persistent $\mathcal{LL}^{(r)}$*

11: Initialize snapshot cache $\mathcal{S}$ as empty
12: **for** $\hat{p} \leftarrow p+1$ **to** $n-r-1$ **do**
13: $\quad$ **while** exit criterion for block $\hat{p}$ not met **do**
14: $\quad\quad$ **for** each minibatch $(x, y)$ with identifier batch_id **do**
15: $\quad\quad\quad$ **if** snapshot $\mathcal{S}[p, \text{batch\_id}]$ exists **then**
16: $\quad\quad\quad\quad$ $z \leftarrow \mathcal{S}[p, \text{batch\_id}]$
17: $\quad\quad\quad$ **else**
18: $\quad\quad\quad\quad$ $z \leftarrow \text{StopGrad}\big(f^{[0:p]}(x)\big)$
19: $\quad\quad\quad\quad$ Optionally store $\mathcal{S}[p, \text{batch\_id}] \leftarrow z$ $\qquad$ ▷ SnapCheck: cache activations
20: $\quad\quad\quad$ $z_{\hat{p}} \leftarrow f_{\hat{p}}(z)$
21: $\quad\quad\quad$ $\tilde{z}_{\hat{p}} \leftarrow A_{\hat{p}}(z_{\hat{p}})$ $\qquad\qquad\qquad$ ▷ Adapter; identity if not needed
22: $\quad\quad\quad$ $\hat{y} \leftarrow \mathcal{LL}^{(r)}(\tilde{z}_{\hat{p}})$
23: $\quad\quad\quad$ $\mathcal{L}_2 \leftarrow \ell(\hat{y}, y)$
24: $\quad\quad\quad$ **Backward/Update:** backprop through $\{f_{\hat{p}}, \mathcal{LL}^{(r)}, A_{\hat{p}}\}$; update only $\{f_{\hat{p}}, \mathcal{LL}^{(r)}, A_{\hat{p}}\}$
25: $\quad\quad$ **Commit and freeze** $f_{\hat{p}}$; **discard** $A_{\hat{p}}$; **extend base** $p \leftarrow \hat{p}$
26: $\quad\quad$ Optionally clear or rebuild $\mathcal{S}$ for the new prefix depth $p$
27: $\quad\quad$ **if** global exit criterion satisfied **then break**
28: **Output:** committed backbone $F_{0:n}$ and $\mathcal{LL}^{(r)}$; inference uses the standard forward $h \circ F_{0:n}$ (adapters and snapshots discarded)

---

## B  ARCHITECTURES AND TRAINING REGIMES: RESNET-18 AND RESNET-50

### B.1  COMMON DATA & OPTIMIZATION SETTINGS

**Dataset and preprocessing.** All experiments use CIFAR-10 with ImageNet-style crops for stronger invariance:

- **Train transforms:** `Resize(256)` $\rightarrow$ `RandomResizedCrop(224, scale=(0.6,1.0))` $\rightarrow$ `RandomHorizontalFlip()` $\rightarrow$ `RandAugment(num_ops=2, magnitude=9)` $\rightarrow$ `ToTensor()`

$\rightarrow$ `Normalize(mean=(0.4914,0.4822,0.4465),`
`std=(0.2023,0.1994,0.2010))` $\rightarrow$ `RandomErasing(p=0.25,`
`scale=(0.02,0.2), ratio=(0.3,3.3))`.

- **Test transforms:** `Resize(256)` $\rightarrow$ `CenterCrop(224)` $\rightarrow$ `ToTensor()` $\rightarrow$ `Normalize` (same statistics).

**Optimization.** Unless specified otherwise, we use **AdamW** with lr $= 3 \times 10^{-4}$, $\beta_1 = 0.9$, $\beta_2 = 0.999$, weight decay $= 0.01$, batch size $= 256$, mixed precision (AMP), cross-entropy loss. E2E uses 400 epochs; LW/SegProp use 400 epochs per stage.

### B.2 RESNET-18

#### B.2.1 BACKBONE SEGMENTATION AND HEAD DEFINITIONS

**Backbone split (224×224 inputs).**

**seg1. seg1:** conv1 $\rightarrow$ bn1 $\rightarrow$ ReLU $\rightarrow$ maxpool $\rightarrow$ layer1 produces $[64, 56, 56]$.

**seg2. seg2:** layer2 produces $[128, 28, 28]$.

**seg3. seg3:** layer3 produces $[256, 14, 14]$.

**seg4. seg4:** layer4[0] (full) + layer4[1].(conv1 $\rightarrow$ bn1 $\rightarrow$ ReLU), producing identity $\mathbf{I} \in \mathbb{R}^{B \times 512 \times 7 \times 7}$ and pre-activation $\mathbf{P} \in \mathbb{R}^{B \times 512 \times 7 \times 7}$.

**Global head (SegProp).** Owns *only* layer4[1].conv2 and bn2 (last conv + BN), then:

$$\hat{\mathbf{Z}} = \text{bn2}(\text{conv2}(\mathbf{P})) + \mathbf{I}, \qquad \hat{\mathbf{Z}} \leftarrow \text{ReLU}(\hat{\mathbf{Z}}).$$

Global average pooling and a two-layer MLP ($512 \rightarrow H \rightarrow 10$, $H = 512$, dropout 0.3) produce the logits.

#### B.2.2 E2E (END-TO-END) TRAINING

- **Structure:** Standard ResNet-18, classifier replaced by MLP $512 \rightarrow 512 \rightarrow 10$.
- **Flow:** Full network forward; full network backprop each epoch.
- **Schedule:** 400 epochs; AdamW (as above); AMP on.

#### B.2.3 LW (LAYER-WISE) TRAINING

**Auxiliaries.** Each stage uses a **Conv–BN–ReLU mapping adapter** to normalize to the canonical head resolution, followed by **GAP + Linear** for local classification:

- Stage 1 aux: $[64, 56, 56] \rightarrow [512, 7, 7]$ via strided convs $\Rightarrow$ GAP $\Rightarrow$ Linear($512\rightarrow10$).
- Stage 2 aux: $[128, 28, 28] \rightarrow [512, 7, 7] \Rightarrow$ GAP $\Rightarrow$ Linear.
- Stage 3 aux: $[256, 14, 14] \rightarrow [512, 7, 7] \Rightarrow$ GAP $\Rightarrow$ Linear.
- Stage 4 aux: input $[512, 7, 7]$ (no mapping) $\Rightarrow$ GAP $\Rightarrow$ Linear.

We use 5 LW stages for ResNet-18: **seg1**, **seg2**, **seg3**, **seg4** (each with an auxiliary head), and a final **fc** stage that trains only the global classifier with the backbone frozen.

**Flow and sizes.**

| Stage | Forward (to aux) | Aux input size |
|---|---|---|
| 1 | seg1$(x)$ | $[64, 56, 56]$ |
| 2 | seg1 $\rightarrow$ seg2 | $[128, 28, 28]$ |
| 3 | seg1 $\rightarrow$ seg2 $\rightarrow$ seg3 | $[256, 14, 14]$ |
| 4 | seg1 $\rightarrow$ seg2 $\rightarrow$ seg3 $\rightarrow$ seg4 | $[512, 7, 7]$ |

**Freezing.** Previous segments are frozen; only the current segment + aux are trainable.

### B.2.4 SEGPROP TRAINING

**Auxiliaries (mapping only).** Aux1/Aux2/Aux3 output *both* $(\mathbf{I}, \mathbf{P})$ in $\mathbb{R}^{B \times 512 \times 7 \times 7}$:

- Aux1: from $[64, 56, 56]$ to $(\mathbf{I}, \mathbf{P})$.
- Aux2: from $[128, 28, 28]$ to $(\mathbf{I}, \mathbf{P})$.
- Aux3: from $[256, 14, 14]$ to $(\mathbf{I}, \mathbf{P})$.

**Flow and sizes.**

| Stage | Forward | Aux output | Head input |
|---|---|---|---|
| 1 | $\text{seg1} \xrightarrow{\text{aux1}}$ | $(\mathbf{I}, \mathbf{P})$ | $[512, 7, 7]$ each |
| 2 | $\text{seg1} \rightarrow \text{seg2} \xrightarrow{\text{aux2}}$ | $(\mathbf{I}, \mathbf{P})$ | $[512, 7, 7]$ each |
| 3 | $\text{seg1} \rightarrow \text{seg2} \rightarrow \text{seg3} \xrightarrow{\text{aux3}}$ | $(\mathbf{I}, \mathbf{P})$ | $[512, 7, 7]$ each |
| 4 | $\text{seg1} \rightarrow \text{seg2} \rightarrow \text{seg3} \rightarrow \text{seg4}$ | $(\mathbf{I}, \mathbf{P})$ | $[512, 7, 7]$ each |

**Freezing.** At each stage, previous segments are frozen. In Stages 1–3 the aux feeds the head directly. Stage 4 trains **seg4 + head**.

### B.3 RESNET-50

We adopt the standard ImageNet-style ResNet-50 ($7 \times 7$ stride-2 stem and max pool; $224 \times 224$ inputs) and view the backbone as four segments $f_1, \ldots, f_4$ producing intermediate representations:

$$z_1 = f_1(x), \quad z_2 = f_2(z_1), \quad z_3 = f_3(z_2), \quad z_4 = f_4(z_3).$$

### B.3.1 BACKBONE SEGMENTATION (COMMON TO ALL HEAD VARIANTS)

**Backbone split ($224 \times 224$ inputs).**

**seg1.** **seg1** ($f_1$)**:** stem + layer1, output $z_1 \in \mathbb{R}^{B \times 256 \times 56 \times 56}$.

**seg2.** **seg2** ($f_2$)**:** layer2, output $z_2 \in \mathbb{R}^{B \times 512 \times 28 \times 28}$.

**seg3.** **seg3** ($f_3$)**:** layer3, output $z_3 \in \mathbb{R}^{B \times 1024 \times 14 \times 14}$.

**seg4.** **seg4 backbone:** the final residual stage layer4, decomposed differently depending on how many convolutions are moved into the global head. We always arrange it so that $f_4$ produces an identity tensor $\mathbf{I}$ from the residual branch and a "pre-head" tensor that serves as input to the global head.

Below we describe three SegProp instantiations that differ in how the last bottleneck layer4[2] is split between the backbone and the global head.

### B.3.2 RESNET-50 E2E TRAINING

- **Structure:** Standard ResNet-50, replacing the ImageNet classifier with an MLP $2048 \rightarrow 512 \rightarrow 10$.
- **Flow:** Full network forward:

$$x \rightarrow f_1 \rightarrow f_2 \rightarrow f_3 \rightarrow f_4 \rightarrow \text{GAP} \rightarrow \text{MLP},$$

and full network backprop each epoch.
- **Schedule:** 400 epochs; AdamW; AMP on.

### B.3.3 RESNET-50 LW TRAINING

**Auxiliaries.** For LW (layer-wise) training, we attach local heads that operate on a canonical $[2048, 7, 7]$ representation. Each stage uses a Conv–BN–ReLU adapter to map the stage output to $[2048, 7, 7]$, followed by GAP + Linear($2048 \to 10$):

- Stage 1 aux: $[256, 56, 56] \to [2048, 7, 7] \to \text{GAP} \to \text{Linear}$.
- Stage 2 aux: $[512, 28, 28] \to [2048, 7, 7] \to \text{GAP} \to \text{Linear}$.
- Stage 3 aux: $[1024, 14, 14] \to [2048, 7, 7] \to \text{GAP} \to \text{Linear}$.
- Stage 4 aux: input $[2048, 7, 7] \to \text{GAP} \to \text{Linear}$.

**Freezing.** At stage $s$, we freeze $f_1, \ldots, f_{s-1}$ and train only $f_s$ and its local aux head.

### B.3.4 RESNET-50 SEGPROP: 1-CONV, 2-CONV, AND 3-CONV GLOBAL HEADS

SegProp augments the backbone with small auxiliary adapters $g_1, g_2, g_3$ that map intermediate features $z_1, z_2, z_3$ to the exact tensors expected by a *shared global head $H$*. At stage $s \in \{1, 2, 3\}$ we use $g_s$ to feed $H$ directly; at stage 4 we use the real backbone output. The difference between our three ResNet-50 SegProp variants lies in how many convolutions of the last bottleneck $\text{layer4}[2]$ are assigned to $H$.

**Notation for the last bottleneck.** We denote the three conv+BN blocks of $\text{layer4}[2]$ by

$$\text{conv1} + \text{bn1}, \qquad \text{conv2} + \text{bn2}, \qquad \text{conv3} + \text{bn3},$$

with standard ResNet-50 widths:

$$\text{conv1} : 1024 \to 512, \quad \text{conv2} : 512 \to 512, \quad \text{conv3} : 512 \to 2048.$$

**(A) 1-Conv global head (conv3+bn3 only).** This variant corresponds to the simplest split: the head owns only the final conv+BN of $\text{layer4}[2]$; the backbone owns the rest of $\text{layer4}$.

**Backbone seg4 (1-conv head).**

- seg4 includes: $\text{layer4}[0]$, $\text{layer4}[1]$, and all of $\text{layer4}[2].(\text{conv1} \to \text{bn1} \to \text{ReLU} \to \text{conv2} \to \text{bn2} \to \text{ReLU})$.
- It produces:
$$\mathbf{I} \in \mathbb{R}^{B \times 2048 \times 7 \times 7}, \qquad \mathbf{P} \in \mathbb{R}^{B \times 512 \times 7 \times 7},$$
where $\mathbf{I}$ is the residual identity (from the skip branch) and $\mathbf{P}$ is the pre-activation input to conv3.

**Global head $H^{(1)}$ (1-conv).** Given $(\mathbf{I}, \mathbf{P})$, the head applies the last conv+BN and the classifier:

$$\mathbf{V} = \text{bn3}(\text{conv3}(\mathbf{P})) \in \mathbb{R}^{B \times 2048 \times 7 \times 7},$$

$$\hat{\mathbf{Z}} = \text{ReLU}(\mathbf{V} + \mathbf{I}), \qquad \mathbf{h} = \text{GAP}(\hat{\mathbf{Z}}) \in \mathbb{R}^{B \times 2048},$$
$$\mathbf{o} = \mathbf{W}_2 \, \sigma(\mathbf{W}_1 \mathbf{h} + \mathbf{b}_1) + \mathbf{b}_2,$$

with MLP $2048 \to 512 \to 10$.

**SegProp auxiliaries for 1-conv head.** For stages $s \in \{1, 2, 3\}$, we define

$$x^{(1)} = z_1 \in \mathbb{R}^{B \times 256 \times 56 \times 56}, \quad x^{(2)} = z_2 \in \mathbb{R}^{B \times 512 \times 28 \times 28}, \quad x^{(3)} = z_3 \in \mathbb{R}^{B \times 1024 \times 14 \times 14}.$$

Each aux is a one-conv adapter:

$$h^{(s)} = \sigma\big(\text{BN}(\text{Conv}_{1 \times 1}^{s_7}(x^{(s)}))\big) \in \mathbb{R}^{B \times 2560 \times 7 \times 7},$$

$$\mathbf{I}_s = h^{(s)}[:, 1:2048, :, :], \qquad \mathbf{P}_s = h^{(s)}[:, 2049:2560, :, :],$$

with strides $s_7 \in \{8, 4, 2\}$ for seg1/2/3 to reach $7 \times 7$. These $(\mathbf{I}_s, \mathbf{P}_s)$ are fed into $H^{(1)}$.

**(B) 2-Conv global head (conv2+bn2 + conv3+bn3).** This variant moves the last two conv blocks of $\text{layer4}[2]$ into the head, leaving only conv1+bn1 in the backbone.

**Backbone seg4 (2-conv head).**

- seg4 includes: $\text{layer4}[0]$, $\text{layer4}[1]$, and $\text{layer4}[2].(\text{conv1} \to \text{bn1} \to \text{ReLU})$.
- It produces:

$$\mathbf{I} \in \mathbb{R}^{B \times 2048 \times 7 \times 7}, \qquad \mathbf{P} \in \mathbb{R}^{B \times 512 \times 7 \times 7},$$

  where $\mathbf{P}$ is now the output of $\text{conv1} + \text{bn1} + \text{ReLU}$ of the last bottleneck.

**Global head $H^{(2)}$ (2-conv).** Given $(\mathbf{I}, \mathbf{P})$, the head applies conv2+bn2 and conv3+bn3, followed by the classifier:

$$\mathbf{U} = \text{ReLU}\big(\text{bn2}(\text{conv2}(\mathbf{P}))\big) \in \mathbb{R}^{B \times 512 \times 7 \times 7},$$

$$\mathbf{V} = \text{bn3}(\text{conv3}(\mathbf{U})) \in \mathbb{R}^{B \times 2048 \times 7 \times 7},$$

$$\hat{\mathbf{Z}} = \text{ReLU}(\mathbf{V} + \mathbf{I}), \qquad \mathbf{h} = \text{GAP}(\hat{\mathbf{Z}}) \in \mathbb{R}^{B \times 2048},$$

$$\mathbf{o} = \mathbf{W}_2\, \sigma(\mathbf{W}_1 \mathbf{h} + \mathbf{b}_1) + \mathbf{b}_2.$$

**SegProp auxiliaries for 2-conv head.** The aux design is identical to the 1-conv case: each $g_s$ maps $x^{(s)}$ to $(\mathbf{I}_s, \mathbf{P}_s)$ with shapes $[2048, 7, 7]$ and $[512, 7, 7]$ via a single $1 \times 1$ conv producing 2560 channels (2048+512) at $7 \times 7$:

$$h^{(s)} = \sigma\big(\text{BN}(\text{Conv}_{1\times1}^{s7}(x^{(s)}))\big) \in \mathbb{R}^{B \times 2560 \times 7 \times 7},$$

$$\mathbf{I}_s = h^{(s)}[:, 1{:}2048, :, :], \qquad \mathbf{P}_s = h^{(s)}[:, 2049{:}2560, :, :].$$

These are fed into $H^{(2)}$.

**(C) 3-Conv global head (conv1+bn1 + conv2+bn2 + conv3+bn3).** In our most "head-heavy" variant, the global head owns *all three* conv blocks of the last bottleneck $\text{layer4}[2]$. The backbone seg4 now includes only $\text{layer4}[0]$ and $\text{layer4}[1]$ (plus the residual path up to the point where $\mathbf{I}$ is defined).

**Backbone seg4 (3-conv head).**

- seg4 includes: $\text{layer4}[0]$ and $\text{layer4}[1]$ (full bottlenecks).
- It produces:

$$\mathbf{I} \in \mathbb{R}^{B \times 2048 \times 7 \times 7}, \qquad \mathbf{F}_{\text{in}} \in \mathbb{R}^{B \times 1024 \times 7 \times 7},$$

  where $\mathbf{F}_{\text{in}}$ is the input to $\text{layer4}[2].\text{conv1}$.

**Global head $H^{(3)}$ (3-conv).** Given $(\mathbf{I}, \mathbf{F}_{\text{in}})$, the head applies all three conv blocks of $\text{layer4}[2]$ and the classifier:

$$\mathbf{U}_1 = \text{ReLU}\big(\text{bn1}(\text{conv1}(\mathbf{F}_{\text{in}}))\big) \in \mathbb{R}^{B \times 512 \times 7 \times 7},$$

$$\mathbf{U}_2 = \text{ReLU}\big(\text{bn2}(\text{conv2}(\mathbf{U}_1))\big) \in \mathbb{R}^{B \times 512 \times 7 \times 7},$$

$$\mathbf{V} = \text{bn3}(\text{conv3}(\mathbf{U}_2)) \in \mathbb{R}^{B \times 2048 \times 7 \times 7},$$

$$\hat{\mathbf{Z}} = \text{ReLU}(\mathbf{V} + \mathbf{I}), \qquad \mathbf{h} = \text{GAP}(\hat{\mathbf{Z}}) \in \mathbb{R}^{B \times 2048},$$

$$\mathbf{o} = \mathbf{W}_2\, \sigma(\mathbf{W}_1 \mathbf{h} + \mathbf{b}_1) + \mathbf{b}_2.$$

**SegProp auxiliaries for 3-conv head.** Now the head input is $(\mathbf{I}, \mathbf{F}_{\text{in}})$ with shapes $[2048, 7, 7]$ and $[1024, 7, 7]$. For stages $s \in \{1, 2, 3\}$ we use one-conv adapters that output $2048 + 1024 = 3072$ channels at $7 \times 7$:

$$h^{(s)} = \sigma\big(\text{BN}(\text{Conv}_{1\times1}^{s7}(x^{(s)}))\big) \in \mathbb{R}^{B \times 3072 \times 7 \times 7},$$

$$\mathbf{I}_s = h^{(s)}[:, 1{:}2048, :, :], \qquad \mathbf{F}_{\text{in}}^{(s)} = h^{(s)}[:, 2049{:}3072, :, :],$$

with strides $s_7 \in \{8, 4, 2\}$ for seg1/2/3. These $(\mathbf{I}_s, \mathbf{F}_{\text{in}}^{(s)})$ are fed into $H^{(3)}$.

**Stage-wise SegProp schedule (all head variants).** For any of the three heads $H^{(k)}$ ($k \in \{1, 2, 3\}$), the SegProp schedule is:

- **Stage 1:** forward $x \to f_1 \to x^{(1)}$, apply aux $g_1$ to get the appropriate pair $(\mathbf{I}_1, \cdot)$, feed to $H^{(k)}$, and optimize $\mathcal{L}(\mathbf{o}^{(1)}, y)$. Trainable: $f_1, g_1, H^{(k)}$; frozen: $f_2, f_3, f_4$ and other aux.

- **Stage 2:** forward $x \to f_1 \to f_2 \to x^{(2)}$, apply $g_2$, feed to $H^{(k)}$. Trainable: $f_2, g_2, H^{(k)}$; frozen: $f_1, f_3, f_4$.

- **Stage 3:** forward $x \to f_1 \to f_2 \to f_3 \to x^{(3)}$, apply $g_3$, feed to $H^{(k)}$. Trainable: $f_3, g_3, H^{(k)}$; frozen: $f_1, f_2, f_4$.

- **Stage 4:** forward along the real backbone, including the appropriate seg4 for that head variant, to obtain the real pair (e.g., $(\mathbf{I}, \mathbf{P})$ or $(\mathbf{I}, \mathbf{F}_{\text{in}})$), feed to $H^{(k)}$, and optimize $\mathcal{L}(\mathbf{o}^{(4)}, y)$. Trainable: $f_4$ and $H^{(k)}$; frozen: $f_1, f_2, f_3$ and all aux.

At all stages, the global head $H^{(k)}$ is shared and updated, while each backbone segment is trained only in its own stage. Auxiliary modules are training-only (Stages 1–3) and are discarded after training.

**Modes and stability.** Frozen modules are set to `eval()` so BatchNorm running statistics do not update; only the current-stage segment, its aux, and the head are in `train()` mode. We use mixed precision (AMP) with unscale→clip→step to avoid occasional FP16 spikes.

### B.4 SCHEDULES AND HYPERPARAMETERS (SUMMARY)

**E2E.** 400 epochs, AdamW ($3\times10^{-4}$, $\beta = (0.9, 0.999)$, wd $= 0.01$), batch 256, AMP on. Classifier heads:

- ResNet-18: MLP $512 \to 512 \to 10$.
- ResNet-50: MLP $2048 \to 512 \to 10$.

**LW.** 5 stages (seg1..seg4 + fc), $\sim$400 epochs per stage; per-stage: *current segment + aux trainable; previous frozen*; aux = **Conv–BN–ReLU adapters** to canonical resolution → GAP + Linear; we report *Test (aux)* per stage and *Test (real)* through the assembled model.

**SegProp.** 4 stages (seg1..seg4); global head *always active*. Stages 1–3: do not execute the next segments beyond the current stage; aux produces the exact head inputs (ResNet-18: $(\mathbf{I}, \mathbf{P})$ with 512 channels; ResNet-50: $(\mathbf{I}, \mathbf{F}_{\text{in}})$ with 2048 + 1024 channels). Stage 4: train **seg4 + head**. Previous segments are frozen each stage. We report *Test (real)* each epoch and *Test (aux)* in Stages 1–3.

### B.5 GPU MEMORY MEASUREMENT PROTOCOL

For all CNN memory measurements (ResNet-18 and ResNet-50), we instrument the training scripts using PyTorch's CUDA memory APIs to report peak allocated and reserved memory per epoch. Specifically, at the beginning of each epoch we call

```
torch.cuda.reset_peak_memory_stats(device)
```

and at the end of the epoch we record

```
peak_alloc    = torch.cuda.max_memory_allocated(device)
peak_reserved = torch.cuda.max_memory_reserved(device)
```

These quantities are reported in bytes by PyTorch; in all tables we convert them to GiB via division by $1024^3$.

**Allocated vs. reserved.** The *allocated* value measures the maximum amount of memory actually used by tensors during the epoch. We therefore treat peak allocated memory as our primary metric for model memory demand. The *reserved* value includes additional memory held by the CUDA caching allocator that may not be immediately released back to the driver; as a result, it can remain higher than strictly necessary or fluctuate across epochs due to allocator behavior. For clarity, we focus on peak allocated memory in the main text and tables, and we report reserved memory only for diagnostic purposes.

**Measurement setup.** All measurements are taken:

- on a single GPU, with CUDA_VISIBLE_DEVICES set to isolate the device,
- with batch size 256, mixed-precision (AMP) enabled, and AdamW optimization,
- using the same data pipeline (transforms, normalization, and CIFAR-10 splits) across E2E, LW, and SegProp.

For E2E runs, we report the steady-state peak allocated memory after the first warm-up epoch, which typically stabilizes for subsequent epochs. For SegProp stages, we report the per-stage peak allocated memory over a single epoch in steady state (once the CUDA allocator has warmed up). All reported values correspond to the maximum of max_memory_allocated over the epoch of interest.

### B.6 AUXILIARY NETWORKS (ALL MODELS & REGIMES)

**Design goal.** Auxiliary modules ("aux") are minimal adapters that align intermediate features with the objective used in each training regime: (i) in **SegProp**, aux modules produce the *exact tensors* consumed by the global head (maintaining a single global objective at every stage); (ii) in **Layer-Wise (LW)**, aux modules produce a canonical feature map that is consumed by a *local* classifier (GAP + Linear), providing a stage-local training signal. To unify code and reduce parameters, we use *one-convolution adapters* with stage-specific strides that directly reach the canonical spatial size (7×7 for ResNets).

#### B.6.1 RESNET-18: AUX DESIGNS

**SegProp (one-conv aux).** Let $x^{(s)}$ denote the output tensor of stage $s$ (seg1/seg2/seg3). We construct a *single* $1 \times 1$ convolution that: (1) downsamples spatially to $7 \times 7$ using stride $s\_7$, and (2) outputs 1024 channels that are split evenly into the required pair for the global head:

$$\underbrace{\mathrm{Aux}^{(s)}(x^{(s)})}_{\in \mathbb{R}^{B \times 1024 \times 7 \times 7}} \xrightarrow{\mathrm{split}} \mathbf{I}, \mathbf{P} \in \mathbb{R}^{B \times 512 \times 7 \times 7}.$$

The mapping is:

$$h = \sigma\big(\mathrm{BN}(\mathrm{Conv}_{1 \times 1}^{s\_7}(x^{(s)}))\big), \quad \mathbf{I} = h[:, 1{:}512, :, :], \quad \mathbf{P} = h[:, 513{:}1024, :, :],$$

where $\sigma$ is ReLU and $s\_7$ is the stride needed to reach $7 \times 7$ from the stage's spatial size:

$$s\_7 = \begin{cases} 8 & \text{for seg1 } (56 \to 7), \\ 4 & \text{for seg2 } (28 \to 7), \\ 2 & \text{for seg3 } (14 \to 7). \end{cases}$$

Stage 4 uses the *real* **seg4** (layer4 pre-head transform), no aux.

**LW (one-conv aux + GAP + Linear).** For stage $s \in \{1, 2, 3\}$, the aux produces a canonical feature $\tilde{z} \in \mathbb{R}^{B \times 512 \times 7 \times 7}$ via a single $1 \times 1$ conv (stride $s\_7$ as above), then applies a local classifier:

$$\tilde{z} = \sigma\big(\mathrm{BN}(\mathrm{Conv}_{1 \times 1}^{s\_7}(x^{(s)}))\big), \quad \hat{\mathbf{y}}^{(s)} = \mathbf{W} \, \mathrm{GAP}(\tilde{z}) + \mathbf{b}.$$

Stage 4 aux: identity mapping to GAP (input already $[512, 7, 7]$). The final "fc" stage trains the real head only.

### B.6.2 RESNET-50: AUX DESIGNS

**SegProp (one-conv aux for three-conv head).** For ResNet-50 SegProp with a three-convolution head, the global head expects an *identity* tensor $\mathbf{I} \in \mathbb{R}^{B \times 2048 \times 7 \times 7}$ and an input tensor $\mathbf{F}_{\text{in}} \in \mathbb{R}^{B \times 1024 \times 7 \times 7}$. The aux adapters are as defined above:

$$h^{(s)} = \sigma\big(\text{BN}(\text{Conv}_{1\times1}^{s\text{-}7}(x^{(s)}))\big) \in \mathbb{R}^{B \times 3072 \times 7 \times 7},$$

$$\mathbf{I}_s = h^{(s)}[:, 1\text{:}2048, :, :], \qquad \mathbf{F}_{\text{in}}^{(s)} = h^{(s)}[:, 2049\text{:}3072, :, :],$$

with $s\text{-}7 \in \{8, 4, 2\}$ for seg1/2/3 as above. Stage 4 uses the real $f_4$ to produce $(\mathbf{I}, \mathbf{F}_{\text{in}})$.

**LW (one-conv aux + GAP + Linear).** For $s \in \{1, 2, 3\}$, a single $1 \times 1$ conv (stride $s\text{-}7$) maps the stage features to the canonical $[2048, 7, 7]$, followed by GAP and a linear classifier:

$$\tilde{z} = \sigma\big(\text{BN}(\text{Conv}_{1\times1}^{s\text{-}7}(x^{(s)}))\big), \quad \hat{\mathbf{y}}^{(s)} = \mathbf{W} \, \text{GAP}(\tilde{z}) + \mathbf{b}, \quad \tilde{z} \in \mathbb{R}^{B \times 2048 \times 7 \times 7}.$$

Stage 4 aux: GAP + Linear on $[2048, 7, 7]$.

**Implementation notes.**

- **Parameterization.** All aux adapters use one $1 \times 1$ convolution (stride set to reach canonical spatial size), followed by BN and ReLU. Weights are He-initialized; biases are zeroed where present.

- **Stability.** To avoid numerical drift when segments are frozen, frozen modules are kept in `eval()` mode so BatchNorm running statistics do not update. We apply AMP and use unscale+clip before optimizer steps to avoid infrequent FP16 spikes.

- **SegProp vs LW.** In SegProp, aux outputs *replace* the role of later segments during early stages by feeding the *global head* directly (ResNets: $(\mathbf{I}, \mathbf{P})$ or $(\mathbf{I}, \mathbf{F}_{\text{in}})$). In LW, aux outputs feed a *local* GAP+Linear classifier with its own loss; the global head is only trained in the final stage.

## C  ARCHITECTURES AND TRAINING REGIMES: LLM FINE-TUNE

This appendix describes the model architectures and the common data and optimization settings used in our experiments. We consider two decoder-only Transformer language models:

- **Llama-3.1-8B-Instruct** (32 decoder layers),
- **Mistral-Nemo-Instruct-2407** (40 decoder layers).

Both models are fine-tuned under three regimes: End-to-End (E2E), Layer-Wise (LW), and Segmented Propagation (SegProp). The codebase is shared across all regimes, and differences arise only from which layers are trainable and how gradients are routed.

### C.1  COMMON DATA AND OPTIMIZATION SETTINGS

**Tasks and Benchmarks.** We focus on two standard benchmarks:

- **MMLU** (Massive Multitask Language Understanding),
- **WinoGrande**.

For both tasks, evaluation is performed using the `lm_eval` harness in few-shot mode:

- **MMLU**: 5-shot, micro batch size 16, batch size 384 (Mistral-Nemo-Instruct-2407),
- **WinoGrande**: 5-shot, micro batch size 8, batch size 384 (Llama-3.1-8B-Instruct).

All reported metrics are obtained using the full fine-tuned model in standard forward mode (no auxiliary modules are used at inference).

**Data Loading and Tokenization.** All fine-tuning and evaluation runs use a unified data pipeline. The key components are:

- **Tokenizer:** we use

  ```
  AutoTokenizer.from_pretrained(model_id,
             trust_remote_code=True).
  ```

- **Chat-style formatting:** for instruction-style data, we rely on `tokenizer.apply_chat_template` to construct the packed input sequence from message lists.

- **Padding and special tokens:**
  - we set `tokenizer.pad_token = tokenizer.eos_token` when the model does not define a pad token,
  - BOS and EOS markers are standardized using constants `BOS`, `EOS`.

- **Sequence statistics:** we compute sequence length statistics (max, min, mean, percentiles) using `compute_stats` and `compute_stats_verbose` to verify that sequence lengths remain within model limits.

**Evaluation Functions.** For MMLU and WinoGrande we use task-specific metric functions integrated into the HuggingFace `Trainer`:

- **MMLU:** we enable a dedicated pipeline via:
  - `configure_mmlu_choice_token_sets(tokenizer)` to define the answer choice tokenization,
  - `set_mmlu_eval_active(True)` to activate MMLU-specific processing,
  - `compute_acc_mmlu` for computing multiple-choice accuracy on the letter options.

- **WinoGrande:** we use `compute_accuracy` over the model's predictions on the two candidate spans.

During training, these functions are passed as `compute_metrics` to the HuggingFace `Trainer`. Logits can be preprocessed by `_preprocess_logits_for_metrics` to reduce memory and speed up metric computation.

**Optimization and Distributed Training.** Unless otherwise stated, all fine-tuning runs share the following optimization settings:

- **Optimizer:** AdamW (via HuggingFace / DeepSpeed)
  - learning rate: `1e-6`,
  - betas: $(0.9, 0.999)$,
  - $\epsilon$: $10^{-8}$,
  - weight decay: configured as "auto" or a small positive value as per the DeepSpeed configuration.

- **Learning rate schedule:** linear decay with warmup:
  - warmup ratio typically set to $3\%$ of total training steps.

- **Gradient clipping:** max gradient norm 1.0.

- **Precision:** bfloat16 is used whenever GPU hardware supports it; otherwise fp16 is used. The helper `detect_gpu_config()` automatically selects between bf16 and fp16.

**DeepSpeed and FSDP Configuration.** We employ DeepSpeed ZeRO-3 and, where applicable, PyTorch FSDP for memory-efficient training of the 8B and 40-layer models:

- **Per-GPU micro-batch size:** `train_micro_batch_size_per_gpu = 16`.

- **Gradient accumulation steps:** `gradient_accumulation_steps = 6`.

- **Effective train batch size:** `train_batch_size = 384` (for a reference GPU count; the driver scripts recompute accumulation steps based on the actual number of devices).

- **Precision:** `bf16.enabled = true`.

- **ZeRO-3 optimization:**
  - `stage = 3`,
  - overlapping communication and computation (`overlap_comm = true`, `reduce_scatter = true`),
  - parameter and optimizer state offload to NVMe at `/opt/dlami/nvme`,
  - `gather_16bit_weights_on_model_save = true`.

- **Gradient checkpointing:**
  - partition activations,
  - contiguous memory optimization,
  - CPU checkpointing enabled for further memory savings.

The training script initializes distributed training as follows:

- the process group is created with `dist.init_process_group(backend="nccl")`,

- each rank sets its CUDA device via `torch.cuda.set_device(local_rank)`,

- the model is loaded with `device_map = "auto"` or an explicit mapping when quantization is used.

**Quantization.**   To reduce memory and allow efficient fine-tuning:

- We optionally load the base model in 4-bit or 8-bit mode using `BitsAndBytesConfig`:
  - 4-bit: `load_in_4bit = True`, with a configurable compute dtype (bf16, fp16, or fp32),
  - 8-bit: `load_in_8bit = True`.

Quantization is applied consistently across E2E, LW, and SegProp regimes so that differences in performance are attributable to the training regime and layer selection, not to changes in low-level optimization.

### C.2   LLAMA-3.1-8B-INSTRUCT ARCHITECTURE

**Backbone Structure.**   We use a HuggingFace-style implementation of Llama-3.1-8B-Instruct with a decoder-only architecture. The model consists of:

- A token embedding matrix `embed_tokens` mapping vocabulary indices to $H$-dimensional vectors.

- A stack of $N = 32$ decoder layers:

$$\texttt{self.layers} = [\text{LlamaDecoderLayer}_0, \ldots, \text{LlamaDecoderLayer}_{31}].$$

- A final RMSNorm `norm`.

- An LM head `lm_head` projecting from the hidden dimension $H$ to the vocabulary size $V$.

Each `LlamaDecoderLayer` contains:

- **Self-attention:** `LlamaAttention` with rotary positional embeddings and grouped key/value heads.

- **MLP:** `LlamaMLP` with a gated activation.

- **Normalization:** two `LlamaRMSNorm` instances, one before attention and one before the MLP.

The forward computation follows:

$$x_0 = \texttt{embed\_tokens}(input\_ids), \tag{C.1}$$
$$x_{i+1} = \text{LlamaDecoderLayer}_i(x_i), \quad i = 0, \dots, 31, \tag{C.2}$$
$$h = \texttt{norm}(x_{32}), \tag{C.3}$$
$$\text{logits} = \texttt{lm\_head}(h). \tag{C.4}$$

**Segmentation and Shared Head.** For SegProp and LW regimes, we conceptually partition the 32 layers into:

- a **base prefix** of early layers (e.g., layers 0 to $p-1$, with $p = 18$ in our main runs),
- a set of **intermediate layers** $p, \dots, N-3$,
- a **global head segment** consisting of the last two decoder layers and the LM head:

$$\text{Head} = (\text{LlamaDecoderLayer}_{N-2}, \text{LlamaDecoderLayer}_{N-1}, \texttt{lm\_head}).$$

This segmentation enables:

- **E2E:** all layers are jointly trainable, no segmentation is enforced.
- **LW:** only one layer (or a small segment) is updated at a time, while earlier layers are frozen.
- **SegProp:** at each step, one intermediate layer is trained jointly with the always-active global head segment, while other layers may be frozen or used in a forward-only capacity.

The implementation uses environment variables read in `LlamaModel` to determine which layers are active:

- `LAYER_IDX_TO_TRAIN` selects the current layer index $i$ to update.
- `OMIT_LAYER_IDXS` and `DEAD_LAYER_IDXS` specify, respectively, temporarily omitted and permanently pruned layers.
- `SEGPROP_MODE` selects among:
  - `"e2e"` (all layers trainable),
  - `"efficient_pruning_alg"` (SegProp stages),
  - `"efficient_pruning_alg_base_model"` (initial reduced model).

At the optimizer level, we use `unfreeze_layers_by_indices()` to set `requires_grad = True` only for:

- the layer given by `LAYER_IDX_TO_TRAIN` (during SegProp/LW stages), and
- the last two layers plus `lm_head`, when we treat them as the shared global head.

All other layers keep `requires_grad = False`, making the training behaviour consistent across E2E, LW, and SegProp while allowing us to isolate the contribution of specific layers.

## C.3 MISTRAL-NEMO-INSTRUCT-2407 ARCHITECTURE

**Backbone Structure.** Mistral-Nemo-Instruct-2407 is a Mistral-style decoder-only Transformer. The architecture is analogous to Llama, with the following components:

- Token embedding layer `embed_tokens`.
- A stack of $N = 40$ decoder layers:

$$\texttt{self.layers} = [\text{MistralDecoderLayer}_0, \dots, \text{MistralDecoderLayer}_{39}].$$

- Final RMSNorm `norm`.
- LM head `lm_head`.

Each `MistralDecoderLayer` includes:

- **Attention:** `MistralAttention`, a multi-head self-attention block with rotary positional embeddings and grouped key/value heads.
- **MLP:** `MistralMLP`.
- **Normalization:** `MistralRMSNorm` layers before attention and MLP.

The overall computation is:

$$x_0 = \texttt{embed\_tokens}(input\_ids), \tag{C.5}$$
$$x_{i+1} = \text{MistralDecoderLayer}_i(x_i), \quad i = 0, \ldots, 39, \tag{C.6}$$
$$h = \texttt{norm}(x_{40}), \tag{C.7}$$
$$\text{logits} = \texttt{lm\_head}(h). \tag{C.8}$$

**Segmentation and Shared Head.** As with Llama, we segment the 40 decoder layers into:

- a **base prefix** (e.g., layers 0 to $p-1$ with $p = 18$),
- **intermediate layers** $p, \ldots, N-3$,
- a **global head segment**:

$$\text{Head} = (\text{MistralDecoderLayer}_{N-2}, \text{MistralDecoderLayer}_{N-1}, \texttt{lm\_head}).$$

The same set of environment variables (`LAYER_IDX_TO_TRAIN`, `OMIT_LAYER_IDXS`, `DEAD_LAYER_IDXS`, `SEGPROP_MODE`) control which layers are active and which regime (E2E, LW, SegProp) is applied.

For MMLU runs, the driver:

- initializes a configuration with:
  - early layers $0 \ldots p-1$ and the last two layers active,
  - intermediate layers in the omit set,
- runs an initial reduced-model fine-tuning on MMLU,
- iteratively activates one additional intermediate layer at a time and fine-tunes it jointly with the shared global head segment,
- evaluates each stage on MMLU with `lm_eval` (5-shot).

At each stage, `unfreeze_layers_by_indices()` is used to unfreeze only the current layer of interest and the global head; all other layers remain frozen. This makes the comparison between regimes consistent and isolates the effect of the SegProp scheduling versus E2E fine-tuning.

