# OpenReview forum: "Unlocking the Power of Layer By Layer Training For LLM"
_ICLR.cc/2026/Conference — Submitted to ICLR 2026_

### Official Review · Reviewer_aU8Y · 2025-10-22

**Soundness:** 2
**Presentation:** 1
**Contribution:** 1
**Rating:** 0
**Confidence:** 4

**Summary:**

The paper investigates layer-by-layer training of LLMs from a fine-tuning perspective. It starts with the information bottleneck/loss of layer-by-layer training (though this is usually investigated from the pretraining perspective) and proposes to strategically reintroduce the final layers. The method has two stages: in the first stage, it jointly tunes the first several and the final several layers; in the second stage, it performs iterative layer-wise tuning of the intermediate Layers. Experiments are conducted on Mistral 7B, with fine-tuning on several downstream tasks.

**Strengths:**

* The paper provides extensive discussion of related work, though it does not yet compare against these works experimentally.

* The proposed method looks simple: it fine-tunes the first several and last several layers together, and then iteratively fine-tunes the intermediate layers.

**Weaknesses:**

* The biggest problem for me is that comparisons between layer-by-layer training and E2E training are usually investigated from a training-from-scratch/pretraining perspective, as reflected in the cited references. From this perspective, layer-by-layer training can be challenging in accuracy but promising in efficiency, offering a new learning paradigm. The paper also seems motivated by the information bottleneck of layer-by-layer pretraining and targets at its failure in transformers. However, it then applies layer-by-layer training in a fine-tuning setting, as shown in the experiments and in the appendix. With today’s well-pretrained LLMs, layer-by-layer fine-tuning is much easier. The paper uses this fine-tuning case to argue that the difficulties of layer-by-layer training can be largely alleviated by the proposed segmented propagation.

* The experiments are not solid. Since the experiments fine-tune an LLM, could the authors compare with other efficient methods such as LoRA, which alleviates the memory burden of end-to-end training? In addition, the results in the main paper are hard to interpret: what is the final performance under given memory/latency constraints? Can the authors report final accuracy together with the corresponding efficiency statistics, given that the proposed method has an efficiency-related hyperparameter?

* The paper discusses the information bottleneck extensively (e.g., the HSIC metric) and claims to build on its principles (e.g., line 082) . However, how do the proposed equations (Eqs. 3.5–3.8) relate to the information bottleneck?

* The writing needs substantial improvement. The structure is unclear. For example, there is at most one page of experiments (half a page of text and half a page of figures), while two pages are devoted to discussion and future work. Also, what is the main difference between Sections 3.5 and 2.4? Both appear to discuss differences between the proposed method and current early-exit and compression methods. Meanwhile, it's better to state the fine-tuning setting clearly in the abstract and introduction to avoid readers mistaking the paper as addressing pretraining challenges of layer-by-layer training.

**Questions:**

Please see above.

---

> ### Author Response · Authors · 2025-11-20
>
> We thank the reviewer for the detailed and constructive feedback. Below we summarize the main changes made in the revised manuscript and how they address your concerns.
>
> 1. Scope: pretraining vs. fine-tuning
>
> We agree that much of prior work on layer-wise (LW) vs. end-to-end (E2E) training focuses on training from scratch, and our original draft did not make our setting explicit enough.
>
> In the revised paper we clarify that:
>
> - Our contributions cover (i) training from scratch for CNNs (ResNet-18/ResNet-50 on CIFAR-10) and (ii) fine-tuning for LLMs (Mistral-Nemo-Instruct-2407 and Llama-3.1-8B-Instruct).
> - The abstract and Section 1 explicitly state this scope, and the beginning of Section 3 explains that SegProp is first defined for a generic deep network and then instantiated for CNNs and Transformers.
> - We soften the claims: SegProp is presented as a simple, practical LW variant that recovers most of the E2E performance under realistic memory limits, not as a full solution to all LW pretraining challenges.
>
> 2. Additional experiments and efficiency reporting
>
> (a) CNNs trained from scratch
>
> To address the concern that we only handle the fine-tuning regime, we added from-scratch CNN experiments:
>
> - Section 4.1 (ResNet-18, CIFAR-10, Fig. 2): LW 93.69%, SegProp 95.23%, E2E 95.50%.
> - Section 4.2 (ResNet-50, CIFAR-10, Fig. 3): LW 90.04%, SegProp 94.34%, E2E 95.53%.
>
> SegProp consistently closes most of the LW–E2E gap in this classical supervised setting.
>
> (b) Memory and efficiency
>
> We now report explicit peak GPU memory:
>
> - Section 4.3 and Table 1 (ResNet-50, batch 256, AMP):
>   E2E: 11.29 GB;
>   SegProp segments: 7.09 / 4.48 / 3.47 / 2.45 GB.
>
> Thus SegProp provides about 1.6x–4.6x lower peak memory while maintaining near-E2E accuracy. Appendix B.5 details the measurement protocol.
>
> (c) LLM final performance and baselines
>
> Section 4.4 and Table 2 summarize final MMLU and Winogrande scores for different base depths.
>
> - For Mistral-Nemo-Instruct-2407: original 67.70 / 82.79, E2E 69.31 / 82.72, SegProp (base 25) 69.38 / 81.37, with base 32 similarly close to E2E.
> - For Llama-3.1-8B-Instruct: SegProp approaches E2E performance over several bases (see Figs. 4–5).
>
> We agree that LoRA/PEFT are important efficient baselines. Due to compute limits we could not include a full LoRA comparison in this revision; Section 5 now explicitly states this limitation and discusses how SegProp and LoRA are complementary and could be combined in future work.
>
> 3. Information bottleneck, HSIC, and relation to our equations
>
> The reviewer asks how the proposed equations relate to the information bottleneck (IB) discussion.
>
> In the revised paper we clarify that:
>
> - IB and HSIC are used as conceptual motivation and diagnostics, not as a formal objective we directly optimize;
> - The main IB-inspired design choice is to keep a persistent global head that provides the same task-level loss to all segments, rather than independent local heads.
> - Added to section 5 limitation - "A more formal information-theoretic analysis, building on HSIC and information-bottleneck tools, may clarify when a single persistent head is sufficient and when richer global supervision pathways are beneficial"
>
> Concretely:
>
> - Section 2.1 ("Layer-wise Training and Information Bottleneck") emphasizes that IB/HSIC explain why purely local LW training tends to lose task-relevant information with depth.
>
> - We explicitly avoid claiming that we are directly solving an IB optimization.
>
> 4. Structure and clarity
>
> We have reorganized and clarified the manuscript:
>
> - Section 2 now cleanly groups related work, including early-exit and compression methods.
> - Section 3 focuses on SegProp (problem setting, two-stage strategy, formalization, SnapCheck)
> - Section 4 is expanded with multiple figures and two tables, covering CNN and LLM experiments as well as memory analysis; captions and fonts were improved for readability.
> - Added" limitations" in section 5
>
> 5. Clarifying that LLM results are for fine-tuning
>
> To avoid confusion about pretraining vs. fine-tuning:
>
> - The abstract and Section 1 explicitly state that the LLM results are in the fine-tuning setting;
> - Section 4.4 states that we start from pretrained Mistral-Nemo-Instruct-2407 and Llama-3.1-8B-Instruct checkpoints and then apply E2E, LW, and SegProp fine-tuning;
> - Appendix C ("Architectures and Training Regimes: LLM fine-tune") provides full details of the fine-tuning setup.
>
> We hope these revisions address your concerns and improve the clarity, scope, and empirical support of the paper.

---

### Official Review · Reviewer_td9j · 2025-10-25

**Soundness:** 3
**Presentation:** 4
**Contribution:** 3
**Rating:** 6
**Confidence:** 3

**Summary:**

The authors introduce a novel method to train LLM layer-wise efficiently. Instead of training layer-per-layer sequentially, this method proposes to first train a reasonable subnetwork, and only then start the sequential layer-per-layer training. At each stage, the same LLM head is reused and participate to the training, instead of ancillary LLM heads, discarded at each step, as in standard LW training.

**Strengths:**

* The method appears to be novel, efficient, and to perform better or as well as E2E training
* The exposition is very clear, which I appreciate

**Weaknesses:**

* The Hilbert-Schmidt Independence Criterion (HSIC) is used to motivate the research as it allows to show that standard LW training suffers from information degradation compared to E2E. However, the paper doesn't compare the new method using this criterion. Either downplaying HSIC in the narrative, or having experiments around this would help.

* Figure 2 would need more detailed explanations (along with bigger fonts and graphs).

* A paragraph describing precisely what is meant by "standard LW" training would be useful.

* A larger set of experiments with similar results would be useful since the proposed method has no theoretical backing. (What about ResNet and image classification? )

* Some theory on why this method should work beyond the datasets used would be helpful (Is the method only working for LLMs? Why should it work better in general?)

**Questions:**

Can you address some of the weaknesses outlined in the "Weaknesses" section?

---

> ### Author Response · Authors · 2025-11-20
>
> Thank you for the thoughtful and constructive review. We have substantially revised the paper to address your comments. Below we respond point by point and reference specific sections in the revised manuscript.
>
> 1) HSIC / information bottleneck vs experiments
>
> - We now clearly position HSIC/IB as conceptual background only.
> - Section 1 (Introduction, paragraphs 2–3) and Section 2.1 state that prior HSIC/IB work explains why purely local layer-wise objectives can discard task-relevant information, and that our contribution is algorithmic and empirical (SegProp), not a new HSIC analysis.
> - Section 5 (Discussion and Future Work) explicitly lists HSIC-based analysis as future work rather than part of this submission.
>
> 2) What is “standard LW” and how does SegProp differ from LW and E2E?
>
> You asked for a precise definition of standard layer-wise training and a clearer contrast with SegProp and E2E.
> Added text in 3.1 "E2E training remains the standard approach: the network runs forward to produce outputs, the loss is computed, and stochastic gradient descent (SGD) applies the chain rule to propagate gradients through \emph{all} layers in reverse order.
>
> LW training, in contrast, proceeds stage-wise. For the first segment, the model runs forward through that segment and a small auxiliary module that aligns its output to the task objective; the segment is optimized against a local loss while the rest of the network remains frozen. Subsequent stages repeat this pattern for each segment: the network runs up to the current segment, a geometry-matching auxiliary module produces the supervision signal, the loss is computed, and only the active segment is updated. Auxiliary modules are training-only and discarded at inference.
>
> We propose Segmented Propagation (SegProp): a method grounded in LW’s segmented optimization but enhanced by reintroducing the final layers during every stage. Concretely, SegProp maintains a \emph{shared global head}—the final few layers that produce the network’s output—throughout training. The active segment is trained against a \emph{single task-level loss} produced by this shared head (fed via lightweight adapters to match geometry when needed). This restores global supervision and aligns representations across stages, helping recover information loss..."
>
> 3) Breadth of experiments: beyond LLMs
>
> - We added a full CIFAR-10 study
>
>  - Section 4.1 (ResNet-18): Figure 2 shows that SegProp reaches 95.23% accuracy vs 93.69% for standard LW and 95.50% for E2E. Panels (b,c) show concatenated LW and SegProp stages.
> - Section 4.2 (ResNet-50): Figure 3 shows SegProp at 94.34%, LW at 90.04%, E2E at 95.53%, plus an ablation of different global-head sizes and stage-by-stage SegProp accuracies (seg1 89.96%, seg2 92.47%, seg3 93.08%, seg4 94.34%).
> - Appendices B.2 and B.3 detail segmentation, auxiliary network and training setup.
> - These results show that SegProp is not LLM-specific and consistently outperforms standard LW while approaching E2E on CNNs.
>
> 4) Why should the method work generally?
>
> Section 5 explains the key intuition:
> - Standard LW tends to fail because each block is optimized only for its local auxiliary head, which can encourage representations that are misaligned with the final task.
> - SegProp keeps a single persistent global head and task loss for all segments, providing “global supervision” while still training one segment at a time.
> - The new ResNet-18/50 experiments empirically support that this mechanism transfers beyond LLMs.
>
> 5) Figures
>
> You commented that figures could be clearer.
>
> - We revised figures and captions:
>    - Figure 2-3 have a clearer LW vs E2E vs SegProp schematic and caption.
>    - Figures 4 and 5 (ResNets) have clearer subcaptions, larger fonts, and captions that state best accuracies.
>    - LLM diagnostic figures clearly label base depths and curve meanings (SegProp variants, LW, Original, E2E).
>
> We hope these changes address your concerns and improve the clarity and scope of the paper. Thank you again for your detailed feedback.

---

> > ### Comment · Reviewer_td9j · 2025-11-26
> >
> > Thank you for your explanations. Given you answer my questions and that the revised paper is significant improvement I am raising my score to an accept.

---

### Official Review · Reviewer_BU3k · 2025-10-30

**Soundness:** 2
**Presentation:** 1
**Contribution:** 1
**Rating:** 2
**Confidence:** 3

**Summary:**

The paper proposes Segmented Propagation (SegProp), a two-stage, layer-by-layer training scheme for LLMs. Stage 1 jointly trains a base prefix with the final layers, and Stage 2 iteratively trains each intermediate layer with final layers. The authors claim SegProp matches or even exceeds end-to-end training on MMLU and HumanEval+, with efficiency benefits.

**Strengths:**

1. Clear, simple idea. Reuse the actual final head/layers as a universal supervisor for each segment, avoiding auxiliary heads and keeping the training objective aligned with the final task.

**Weaknesses:**

1. The major issue with this paper is its very insufficient experiments (only Figure 2 is for experiments).  The author claims that “matches/exceeds E2E training”. However, experiments appear limited in model settings/baselines and do not compare against strong modern alternatives for local or modular training (deep supervision, synthetic gradients, etc.). Claims of exceeding E2E need broader, compute-normalized comparisons.
2. Key training details are missing. Early-exit during training is mentioned conceptually but concrete criteria aren’t specified, which affects reproducibility.
3. Efficiency claims lack hard numbers. The paper argues reduced recomputation and energy use via SnapCheck, but provides no quantitative profiling of peak memory, throughput, FLOPs, or energy under fixed hardware/parallelism.
4. Results presentation needs tightening. It is more like a undergraduate project report than a research paper. Both writing and plots need polishment.

**Questions:**

N/A

---

> ### Author Response · Authors · 2025-11-20
>
> Thank you for the careful review and helpful suggestions. Below I respond to your main concerns and point to concrete changes in the revised paper.
>
> - 1) “Insufficient experiments / ‘matches/exceeds E2E’ claims need broader comparisons”
>
> We agree the original experiments were too limited. The revision substantially expands them:
>
> - Added CNN training-from-scratch:
>    - ResNet-18 and ResNet-50 on CIFAR-10.
>    - Direct comparison of:
>        - End-to-End (E2E),
>        - Standard Layer-Wise (LW) with auxiliary heads,
>        - SegProp.
>    - See:
>       - Section 4.1 “SegProp Training for ResNet-18 on CIFAR-10”
>       - Section 4.2 “SegProp Training for ResNet-50 on CIFAR-10”
>       - Figures 2 and 3.
>
> - Key results (Sections 4.1–4.2):
>    - ResNet-18:
>       - E2E: 95.5%
>       - LW: 93.7%
>       - SegProp: 95.2%
>    - ResNet-50:
>       - E2E: 95.5%
>       - LW: 90.0%
>       - SegProp: 94.3%
>
> - Expanded LLM fine-tuning:
>    - Section 4.4 “SegProp Fine-Tuning for LLMs”
>       - Figures 4–5 and Table 2.
>       - Llama3.1-8B-Instruct, Winogrande (5-shot):
>          - E2E: 79.1%
>          - SegProp (Base 21): 78.9%
>
> We also softened the claims: the Abstract and Conclusion now state that SegProp “matches or closely approaches E2E” under the reported settings, rather than claiming broad superiority.
>
> - 2) “Key training details / early-exit criteria missing”
>
> You are right.
>
> - Clarified role of early exit:
>    - We removed most of the text related to early exit. The main empirical focus is on performance under segmented training.
>    - ResNets (Appendix B.4, “Schedules and Hyperparameters (Summary)”):
>       - Fixed 400 epochs per stage; no early stopping, to ensure fair E2E/LW/SegProp comparison.
>    - LLMs (Appendix C.1–C.3):
>       - Fixed training budget of 1 epoch.
>
> Added full Appendix B and C detailing training settings.
>
> - 3) “Efficiency claims lack hard numbers (SnapCheck, memory, energy)”
>
> We added concrete memory measurements. We removed unsupported claims from Snapshot text. Keeping "Snapshot checkpointing (SnapCheck). Complementary to AC/SAC, SnapCheck caches detached activations from the frozen prefix and reuses them across iterations, reducing redundant forward compute without increasing peak memory. It integrates naturally with SegProp and is used
> only during training."
>
> - New memory profiling (Section 4.3, Table 1)
> - ResNet-50 on CIFAR-10, batch 256, AMP, single GPU.
>    - Peak allocated memory from torch.cuda.max_memory_allocated:
>       - E2E (full model): 11.29 GB
>       - SegProp:
>          - seg1: 7.09 GB
>          - seg2: 4.48 GB
>          - seg3: 3.47 GB
>          - seg4: 2.45 GB
>       - So SegProp reduces peak memory by roughly 1.6×–4.6×, depending on stage.
>
> - 4) “Results look like a project report; writing and plots need polishing”
>
> We restructured and polished both content and figures.
>
> - Structure:
>    - Sections 1–3: Motivation, related work, and method (with formalization in Section 3.3).
>    - Section 4: Expanded “Results and Analysis”, with:
>       - 4.1 ResNet-18,
>       - 4.2 ResNet-50,
>       - 4.3 Memory comparison,
>       - 4.4 LLM fine-tuning.
>    - Section 5: “Discussion and Future Work” including limitations.
>    - Appendices:
>       - A: Algorithm (SegProp-SGD),
>       - B: Detailed CNN architectures/training regimes,
>       - C: Detailed LLM architectures/training regimes.
>
> - Figures:
>    - Larger fonts, simplified legends.
>    - More descriptive captions:
>     - Fig. 2: ResNet-18 E2E vs LW vs SegProp, stage-concatenated learning curves, best-epoch dots.
>     - Fig. 3: ResNet-50 E2E vs LW vs SegProp, plus head-size ablation.
>     - Figs. 4–5: clear explanation of curves (SegProp variants, LW, Original, E2E) and x-axis (layer index vs lm head).
>
> We appreciate your feedback—it directly motivated the broader experiments, the concrete memory profiling, and clearer presentation.

---

### Official Review · Reviewer_ftzy · 2025-10-31

**Soundness:** 3
**Presentation:** 3
**Contribution:** 3
**Rating:** 6
**Confidence:** 3

**Summary:**

This paper studies layer-wise training of multi-layer transformers similar to as is done for deep neural networks. They propose such a training paradigm SegProp, where a few layers are first trained, after which those weights are frozen and the remaining layers are trained. The paper also studies early stopping, and the impact it has on training.

**Strengths:**

- This is an important topic. The paper that gives a training paradigm for transformers that is similar to something that has often been used for deep neural networks.
- The proposed algorithm is simple and it does work well empirically.
- The analysis of layer-wise training, early stopping and other related ideas are explained in details.
- The ideas are presented very lucidly and the paper is easy to read.

**Weaknesses:**

- It would be good to highlight how this work is different from pure LW training, E2E training etc.
- Please explain what kind of convergence criterion/exit criterion is usually used.
- Line 195: Please define f_{embed} and f_{LM} separately. Also in the definition of f_i(x) in line 197, it is better to keep an i-subscript f_{MLP_i}, f_{SA_i} to show that it corresponds to the i-th layer.
- Also in definitions, the functions’ domain and range needs to be defined, like F: R^{d_1} -> R^{d_2}. Same for LL.

**Questions:**

- Line 103: What is global supervision?
- Line 225: Is the model depth p a constant or a hyperparameter? How is it chosen?

---

> ### Author Response · Authors · 2025-11-20
>
> We thank the reviewer for the helpful feedback and the positive assessment of our contribution and clarity. Below we address the main points and indicate where changes were made.
>
> 1) Difference from pure LW and E2E training
> We clarified how SegProp differs from standard layer‑wise (LW) and end‑to‑end (E2E) training, both conceptually and experimentally.
>
> Sections 1 and 3.1, together with Fig. 1, now explicitly contrast:
> (i) standard LW, where each segment is trained with its own auxiliary head and local loss;
> (ii) E2E training, which uses a single global loss over the full depth; and
> (iii) SegProp, which trains one segment at a time but always under a single task‑level loss computed by a persistent global head (last layers + LM/classification head). Section 3.1 (“Problem Setting”) explains how SegProp is “grounded in LW training but enhanced by reintroducing the final layers during every stage” while still differing from full E2E backprop.
>
> Appendix B.2–B.3 now give a precise definition of our “standard LW” baselines for ResNet‑18/50 (segment structure, auxiliary heads, and freezing policy), which we compare directly against SegProp and E2E in Sections 4.1–4.2.
>
> 2) Exit / convergence criteria and early exit in segmented learning
> We made our exit criteria explicit and clarified how early exit works in the segmented setting.
>
> For the CNN experiments we use a fixed number of epochs per stage (400, same as the E2E schedule), as stated in Sections 4.1–4.2 and Appendix B.2–B.3. For the LLM experiments, we use one epoch per segment with fixed hyperparameters, as described in Appendix C.
>
> In the context of segmented learning, “early exit during training” means that we can stop adding or training further segments once the current segment reaches an acceptable loss/score on a validation set.
>
> 3) Notation for model components and SegProp modules
> We clarified the notation for the model and SegProp components.
>
> Section 3.3 now gives an architecture‑agnostic formulation of the model as a composition of blocks and a task head, and introduces the notation F_{a:b}, f^{[0:p]}, and LL^{(r)} consistently. The text explains how these objects are used in the SegProp update rules and how they relate to the base prefix and last‑layers module, so that the roles of the embedding, head, and layer‑indexed blocks are clearer.
>
> 4) “Global supervision”
> We defined “global supervision” more precisely.
>
> Sections 1, 2.1, and 3.3 now emphasize that standard LW “lacks global supervision” because each segment is trained to satisfy its own auxiliary head, while SegProp maintains a single global task loss via a persistent global head LL^{(r)} that is active in every stage. In other words, “global supervision” means that all segments are optimized against the same end‑task loss produced by the final layers + head, rather than independent local objectives.
>
> 5) Is p (model depth) a constant or a hyperparameter?
> We clarified that p is a hyperparameter and explained how it is chosen.
>
> Section 3.3 states that we choose a base prefix depth p and a last‑layers size r, making it clear that p is not fixed by the architecture.
>
> Section 4.4 and Table 2 explore multiple base depths (e.g., p=1,18,25,32 for Mistral‑Nemo and p=1,18,21,24 for Llama‑3.1‑8B), and we discuss how an appropriate p is selected based on these empirical trade‑offs.
>
> We hope these changes address your concerns and further clarify the contribution and technical details of SegProp.

---

### Meta-Review · Area_Chair_aWx1 · 2025-12-22

**Summary:**

This paper proposes SegProp, a segmented layer-wise training procedure intended to reduce memory and optimization difficulties of end-to-end training while maintaining comparable performance. The paper’s strongest motivation is about the challenges of training from scratch, while a substantial portion of the original evidence is on LLM fine-tuning, where the baseline landscape and constraints differ substantially. The rebuttal has added some pre-training comparisons, yet remains limited in terms of the baselines and discussions on the performance and efficiency tradeoffs. It is in general unclear when SegProp will be preferred to other more recent parameter and memory efficient baselines for training and fine-tuning.

**Reviewer Concerns:**

Concerns on clarity of definitions and training protocol, empirical efficiency measures, more evidence apart from LLM fine-tuning have been addressed in rebuttal. The main concerns on incomplete baselines, positioning of the paper (training versus fine-tuning) remain valid.

**Reviewer Scores:**

The two reviewers mostly asked for clarifications and extensive experiment settings, which are likely to remain borderline and become positive. The two reviewers with negative ratings may remain negative after the rebuttal since their main concerns are not fully addressed.

---

### Decision · Program_Chairs · 2026-01-26

Reject